# Controlling protein stability with SULI, a highly sensitive tag for stabilization upon light induction

Miaowei Mao [1,2,3,4], Yajie Qian[1,2,4], Wenyao Zhang[1,4], Siyu Zhou[1,2], Zefeng Wang [3], Xianjun Chen [1,2] ✉ & Yi Yang [1,2] ✉

Optogenetics tools for precise temporal and spatial control of protein abundance are valuable in studying diverse complex biological processes. In the present study, we engineer a monomeric tag of stabilization upon light induction (SULI) for yeast and *zebrafish* based on a single light-oxygen-voltage domain from *Neurospora crassa*. Proteins of interest fused with SULI are stable upon light illumination but are readily degraded after transfer to dark conditions. SULI shows a high dynamic range and a high tolerance to fusion at different positions of the target protein. Further studies reveal that SULI-mediated degradation occurs through a lysine ubiquitination-independent proteasome pathway. We demonstrate the usefulness of SULI in controlling the cell cycle in yeast and regulating protein stability in *zebrafish*, respectively. Overall, our data indicate that SULI is a simple and robust tool to quantitatively and spatiotemporally modulate protein levels for biotechnological or biomedical applications.

Gene editing[1–4], transcriptional regulation[5], and RNA interference[6] are widely used methods to manipulate the level of a protein in order to study its role in complex biological processes. In particular, the CRISPR-Cas system is being championed as a flexible and robust tool for genome editing[7]. These techniques exert their control at the level of genes or messenger RNA (mRNA); thus, there are significant delays before the phenotypes occur due to the inherent turnover of proteins and their mRNA. Endogenous cellular proteins may be acutely depleted by proteasome targeting in the presence of proteolysis-targeting chimeric molecules (PROTACs)[8] or by degradation of specific antibodies and antibody receptors (with Trim-Away)[9]. However, such protein-specific reagents may need extensive development, and the depletion is not readily reversible or tunable. Alternatively, proteins may be depleted conditionally using chemically-induced degrons or stabilization tags, such as SURF and AID[10,11]. These

degrons are rapidly degraded along with their fusion partners in the presence of small molecule ligands, including rapamycin and its derivatives[10,12,13], and auxin[11]. When the inducers are removed, the pool of the protein rebuilds through de novo protein synthesis. Therefore, the stability of the target protein can be controlled directly and reversibly. Nevertheless, as these inducers diffuse freely and are hard to remove from a living system, it is rather challenging to precisely upregulate or downregulate protein levels at an exact time and location.

To this end, light can fulfill almost all the requirements of an ideal trigger for protein quantity control due to its precise spatial and temporal resolution. To date, a few light-induced degrons (LIDs) have been engineered by others and us[14–16] and have been used for light-induced and proteasome-mediated degradation in yeast[15,17], *C. elegans*[18], zebrafish[19], and mammalian cells[14,19]. All these LIDs share a

[1]Optogenetics & Synthetic Biology Interdisciplinary Research Center, State Key Laboratory of Bioreactor Engineering, Shanghai Collaborative Innovation Center for Biomanufacturing Technology, East China University of Science and Technology, 130 Mei Long Road, Shanghai 200237, China. [2]Shanghai Frontiers Science Center of Optogenetic Techniques for Cell Metabolism, School of Pharmacy, East China University of Science and Technology, 130 Mei Long Road, Shanghai 200237, China. [3]Bio-Med Big Data Center, CAS Key Laboratory of Computational Biology, CAS Center for Excellence in Molecular Cell Science, Shanghai Institute of Nutrition and Health, University of Chinese Academy of Sciences, Chinese Academy of Sciences, Shanghai 200031, China. [4]These authors contributed equally: Miaowei Mao, Yajie Qian, Wenyao Zhang. ✉e-mail: xianjunchen@ecust.edu.cn; yiyang@ecust.edu.cn

similar configuration, which consists of an N-terminal blue light-sensitive light-oxygen-voltage (LOV) domain from higher plant phototropins and a C-terminal small peptide degron. Under dark conditions, the C-terminal alpha helix of LOV interacts with its core domain, thus sequestering the degron away from the cellular quality control system and rendering it cryptic. LIDs are destabilized upon exposure to blue light, as the C-terminal helix dissociates from the LOV core domain, thus releasing the degron and inducing degradation of the fusion protein through the processive activity of the proteasome. Nevertheless, an inherent problem of these LIDs is the basic level of degron activity that destabilizes the target protein even in the dark and thus attenuates possible dynamic ranges, i.e., the On/Off switching ratio[17]. In addition, the degrons of LIDs, e.g., the C-terminal sequence of ornithine decarboxylase (ODC)[15–17,20] and the small four-amino acid peptide RRRG[14], only work at the C-terminus of a protein; thus, LIDs must be fused at the C-terminus of the target protein, which may prevent their broad application for proteins that do not tolerate C-terminal fusions.

We reason that protein levels may be able to be alternatively controlled by a tag that causes stabilization upon light induction (SULI). Proteins fused to the SULI tag are degraded under dark conditions but stabilized upon light illumination, which is a light-activated behavior opposite that of LIDs. Thus, SULI can be used for direct induction of protein expression upon illumination. Very recently, Hepp et al. developed such a tool based on the LOV domain of *Phaeodactylum tricornutum aureochrome* 1a (AuLOV) and the degron cODC. However, it suffered from a low On/Off switching ratio (approximately twofold) and multimeric nature (exhibiting a dimeric state upon blue light illumination)[21], making it almost unusable in most biological studies. Therefore, SULI, harboring favorable light-inducible characteristics is still highly desirable.

In the present study, we make a discovery involving the light-induced stabilization effects of VVD, the smallest but highly photosensitive LOV domain that has been reported to participate in the blue light response and modulate the gating of the circadian clock in *Neurospora crassa*[22]. We explore the light-induced stabilization effects of VVD and develop a practical and sensitive SULI, which exhibits a combination of favorable light-inducible characteristics, including a high dynamic range and a high tolerance to fusion at different positions of the target protein. Studies reveal that SULI mediates degradation through a lysine ubiquitination-independent proteasome pathway, in which the Hsp104 protein plays an essential role. We demonstrate the usefulness of SULI in regulating the cell cycle and growth of yeast as well as in controlling the embryonic development of zebrafish. Taken together, the findings of the present study indicate that SULI provides a powerful and convenient tool for the study of protein metabolism and protein functions in diverse cellular processes.

## Results

### Design, optimization, and validation of a monomeric photosensitive degron

We previously utilized VVD's remarkable property of blue light-induced dimerization to establish light-switchable gene expression systems, which allowed simple, precise, robust, and highly efficient control of gene expression in bacteria, yeast, or mammalian cells[23–27]. During these studies, we unexpectedly found that the expression level of the soluble VVD fusion protein under a constitutive promoter was significantly enhanced under light conditions, whereas such an effect was not observed for a series of other LOV domains (Supplementary Fig. 1). Further studies showed that blue light had little effect on the mRNA level of the VVD fusion protein (Supplementary Fig. 2), suggesting that the light effect was post-transcriptional. We, therefore, wondered whether blue light was capable of stabilizing the VVD protein in cells and whether this property of VVD could be further

exploited to develop a light-switchable stabilization tag for conditional control of protein degradation (Fig. 1a).

To augment the effects of light-induced stabilization, we carried out mutagenesis of VVD, focusing on the site previously reported to modulate the dimerization or thermal reversion properties of the VVD domain[28,29], and fused it to the mCherry reporter at its C-terminus. Interestingly, only three sites (Y50W, N56K, or C71V) capable of enhancing the stability of light-induced VVD dimerization exhibited more significant differences (~15-fold) in protein expression levels between light and dark conditions than the wild-type VVD (Supplementary Fig. 3). We then performed combinatorial mutagenesis of the three sites and found that the VVD variant simultaneously harboring the mutations had up to an ~30-fold difference in protein expression level between light and dark conditions (Fig. 1b, Supplementary Fig. 4, 5a, b). Similar results were observed when a known highly stable protein, superfolder GFP (sfGFP), was used as the reporter (Supplementary Fig. 6). We further measured the degradation kinetics of the VVD-bearing fusion protein using a cycloheximide (CHX) chase experiment in which protein synthesis was inhibited by CHX when the cells were transferred to dark conditions. The results showed that the half-life of the triple mutant was significantly lower than that of wild-type VVD under dark conditions, whereas both proteins were rather stable under light conditions (Supplementary Fig. 7), suggesting that the enhanced effects of light-inducible stabilization were due to the accelerated degradation of the dark-state VVD mutant in the cells.

Similar to other protein tags, such as fluorescent proteins, an ideal stabilization tag should be monomeric, as oligomerization of the tag may interfere with the functions of its fusion partner. VVD is monomeric in the resting state and dimerizes via symmetric contact of an exposed hydrophobic face upon light illumination[22,30]. This property of VVD has been explored to engineer light-switchable transcription activators or repressors[23–27] and protein dimerization agents[31,32] by us and others. However, it is not readily clear whether there is any correlation between dimerization and stabilization of light-activated VVD protein, in particular, several mutations in SULI are reported to promote the stability of the light-induced VVD dimer[29]. We, therefore, introduced mutations at Tyr40 in the VVD domain that completely abrogate dimerization by disrupting the hydrogen bonds at the dimer interface[33] and analyzed the protein levels of the mutants in cells illuminated by blue light or under dark conditions using mCherry as the reporter. Interestingly, all variants exhibited significant light-induced stabilization of the reporter protein (Fig. 1c, Supplementary Fig. 5c), suggesting that the effect of stabilization and dimerization was uncoupled. Among these variants, the variant with the Y40E mutation showed the largest On/Off switching ratio of stabilization (Fig. 1c, d). Further size-exclusion chromatography data showed that the variant with the Y40E mutation was a monomer in either the dark state or the light-activated state (Supplementary Fig. 8). We, therefore, termed this variant SULI (stabilization upon light induction). Similar light-induced stabilization effects of SULI were also observed using sfGFP and NanoLuc (NLuc) luciferase as the reporters (Fig. 1e, f), demonstrating good adaptability of SULI to different target proteins. Taken together, the results revealed that we obtained a high-performance photosensitive degron that is stable upon light illumination but undergoes fast degradation under dark conditions, which provides a valuable tool to control protein abundance at the posttranslational level.

### SULI has high tolerance to fusion at different positions of the target protein

The previous photoswitchable degrons, e.g., psd3 and AuLOV-cODC1[17,21], consist of an N-terminal blue light-sensitive LOV domain and a C-terminal small peptide degron. Light-induced sequestering or release of the degron leads to stabilization or degradation of the fusion protein. Therefore, these photoswitchable degrons must be fused at

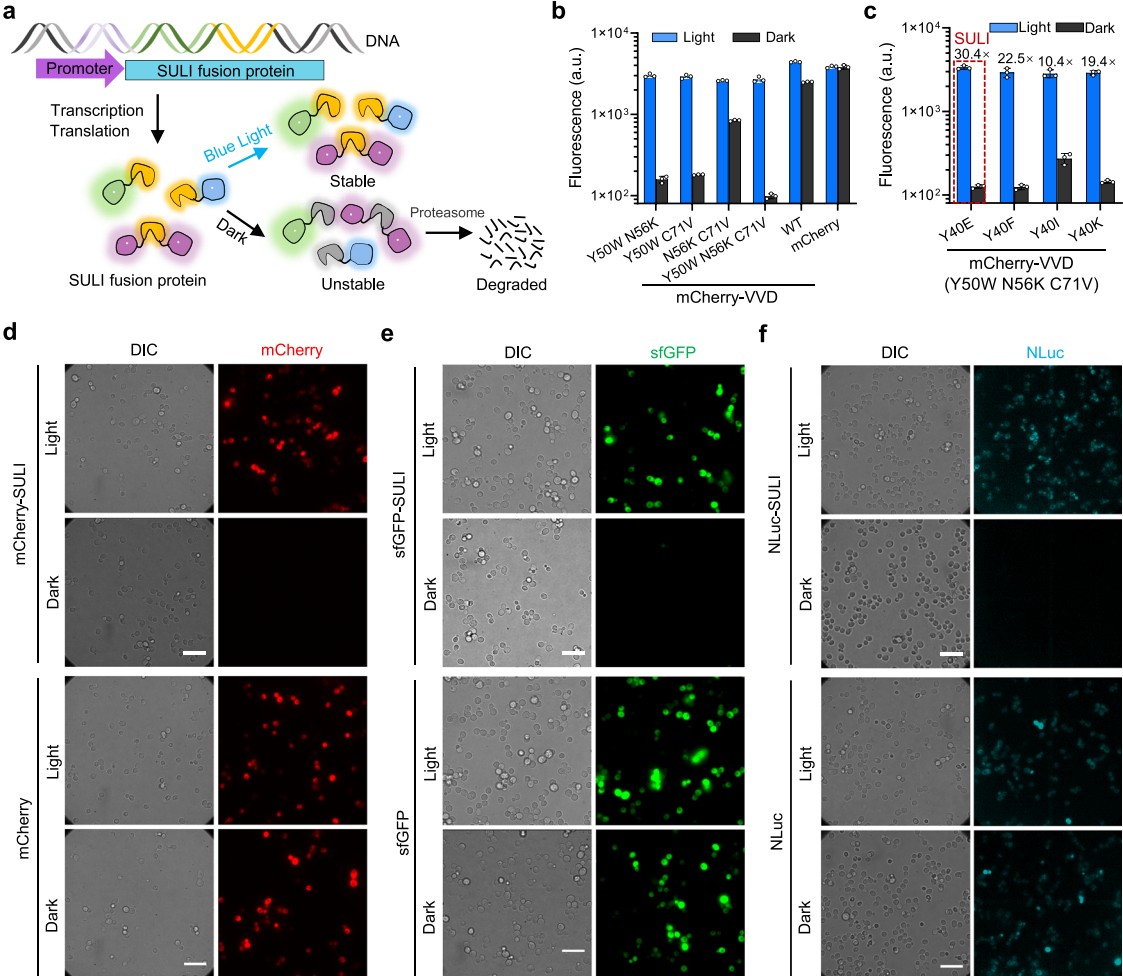

**Fig. 1 | Design, optimization, and validation of a monomeric photosensitive degron. a** Schematic of the SULI-mediated degradation of a protein of interest (POI) by light. The SULI fusion protein is stable upon exposure to blue light but is unstable and degraded by the proteasome under dark conditions. **b** Changes in stabilities between dark and light conditions for different VVD variants. Yeast cells expressing different mCherry-VVD variants were cultured under light or dark condition for 15 h before mCherry fluorescence was analyzed by flow cytometry. The data are presented as the mean ± SD from three biological replicates. a.u., arbitrary units. **c** Effects of the Y40 mutation in VVD (Y50W N56K C71V) on light-induced stabilization. Yeast cells expressing different mCherry-VVD variants containing Y40 mutation were cultured under light or dark conditions for 15 h before mCherry fluorescence was analyzed by flow cytometry. a.u., arbitrary units. The data are presented as the mean ± SD from three biological replicates. **d** Imaging of SULI-mediated degradation of mCherry (**d**), sfGFP (**e**), and NLuc (**f**). Yeast cells expressing different SULI fusions were cultured under light or dark condition for 15 h before fluorescence or luminescence was analyzed by imaging. The scale bars in **d**–**f** are 20 μm. Source data are provided as a Source Data file.

the C-terminus of the target protein to prevent the released degron from being buried in the fusion protein and thus failing to be recognized by the cellular degradation machinery. In comparison, SULI consists of only the VVD domain and does not contain a degron. We, therefore, hypothesized that SULI should have high tolerance to fusion at different positions of the target protein. To test this hypothesis, we fused SULI to the N-terminus, C-terminus or inner position of the reporter protein (Fig. 2a, c, e) and tested its ability to control the stability of the fusion proteins under light and dark conditions. The results showed that SULI-mediated significant light-induced stabilization of the reporter protein for all three configurations (Fig. 2a–f), whereas AuLOV-cODC1 and psd3 only showed light-induced stabilization and destabilization for their C-terminal fusions, respectively (Fig. 2a–f). Notably, the On/Off switching ratio of SULI (~30-fold) was significantly higher than that of AuLOV-cODC1 (~3-fold) (Fig. 2a, b), the only other reported tag for stabilization upon light induction. Furthermore, the protein level of SULI fusion under light conditions was >5-fold that of AuLOV-cODC1 (Fig. 2a, b), revealing that SULI is more favorable for applications in which high levels of proteins are desired to exert their biological functions.

## SULI mediates quantitative and spatiotemporal control of protein stability in yeast cells

We sought to investigate the degradation kinetics of SULI-mediated protein stability using a CHX chase experiment. The engineered cells cultured under light illumination were incubated with CHX and transferred to dark conditions or kept in light conditions. The results showed that mCherry-SULI was degraded after the cells were transferred to the dark, with a half-life of 2–3 h compared to the >10 h under light conditions (Fig. 3a). Similar degradation kinetics were observed for the fusion proteins with C-terminal and inner insertion of SULI (Fig. 3a). To accelerate the degradation kinetics, we also engineered a SULI variant with a significantly shortened half-life of 0.5–1 h under dark conditions, SULI$_f$, by introducing the I74V/C76V/M135L triple mutations in the SULI domain, which are known to affect thermal reversion of VVD (Supplementary Fig. 9, Supplementary Table 1). In addition, SULI could be used to switch the abundance of the target protein in an oscillatory way by regulating the accumulation and degradation repeatedly (Fig. 3b).

We next tested the response of SULI to light intensity by illuminating the cells expressing mCherry-SULI with a gradient of light

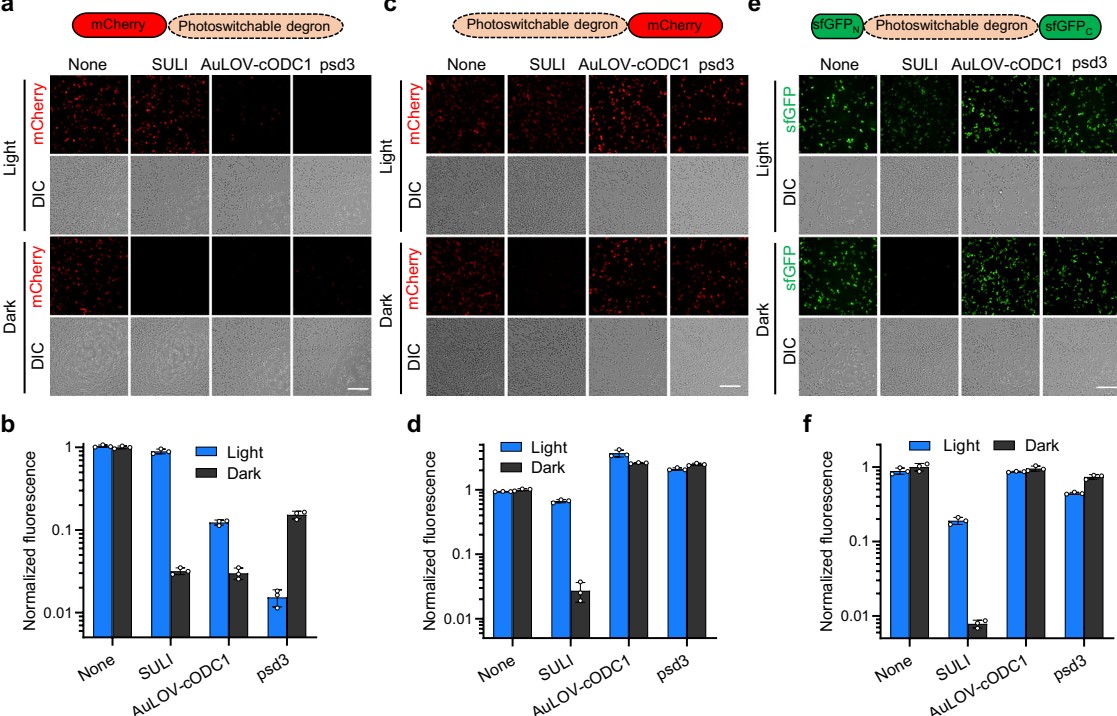

**Fig. 2 | Tolerance of SULI to fusion at different positions of the target protein.** The photoswitchable degrons SULI, AuLOV-cODC1, and psd3 were fused to the C- (**a**, **b**) or N-terminal (**c**, **d**) or the inner region (**e**, **f**) of the reporter protein. Yeast cells expressing these fusion proteins were cultured upon blue light illumination or under dark conditions for 15 h before the fluorescence was imaged by fluorescence microscopy (**a**, **c**, **e**) and quantified by a microplate reader (**b**, **d**, **f**). The fluorescence was normalized to that of the cells expressing reporters without fusion with the photoswitchable degrons under dark conditions. The scale bars in **a**, **c**, and **e** are 50 μm. The data in **b**, **d**, and **f** are presented as the mean ± SD from three biological replicates. Source data are provided as a Source Data file.

irradiance. We observed that the mCherry fluorescence increased in a dose-dependent manner with increasing irradiance of blue light (Fig. 3c). The response of SULI was well-fitted by a Hill function with light sensitivity (half-maximal response, $k$) of 0.78 μmol m$^{-1}$ s$^{-1}$ (Fig. 3c), reflecting higher sensitivity of SULI than of the AuLOV-cODC1 tag (half-maximal response of 4.74 μmol m$^{-1}$ s$^{-1}$). To spatially control protein depletion, yeast cells expressing mCherry-SULI were grown on a solid medium and irradiated by blue light using a mask with a specific image. The mCherry fluorescence image of the cells replicated the pattern of the original image used as the mask (Fig. 3d). These data demonstrate that SULI is a robust tool for quantitative and spatio-temporal control of protein stability in yeast cells.

## SULI mediates protein degradation through a lysine ubiquitination-independent proteasome pathway

In eukaryotes, proteins are degraded mainly through vacuolar-lysosomal pathways (in the vacuolar system in plants and yeast and in lysosomes in mammals) or ubiquitin-proteasome pathways[34–36]. To study the mechanism of SULI-mediated degradation, yeast cells bearing the *erg6* mutation (erg6 mutation enables cells to be hypersensitive to multiple inhibitors, e.g., CHX and MG132[37]) and expressing mCherry-SULI were treated with the proteasome inhibitor MG132 or the vacuolar protease inhibitor phenylmethylsulfonyl fluoride (PMSF). We found that MG132 significantly upregulated the cellular protein levels of mCherry-SULI in darkness, while PMSF had negligible effects (Fig. 4a), suggesting that the proteasome may have been responsible for the degradation of the SULI tag in the cells. It should be noted that MG132 could not completely rescue SULI-mediated degradation under dark conditions even at a high concentration (Fig. 4b), probably because it was difficult for MG132 to diffuse into the cells due to poor permeation of the yeast membrane, which was consistent with previous studies[38]. In comparison, PMSF may have better membrane permeability and has been extensively used in numerous living yeast studies[39,40], but further detection of the status of vacuole upon PMSF treatment will be helpful to bolster the conclusion that vacuolar-lysosomal pathways are not involved in SULI-mediated degradation.

We next sought to investigate whether ubiquitination was involved in proteasome-dependent degradation by SULI. To this end, we created single or multiple mutations of lysine to arginine in SULI to inhibit the ubiquitination of SULI, and tested the capacity of the resulting SULI to mediate the degradation of target proteins under dark conditions. Unexpectedly, the results showed that all these mutants maintained light-inducible stabilization of the mCherry reporter (Fig. 4c, Supplementary Fig. 10). Immunoblot analysis showed that no marked ubiquitinated bands were observed for mCherry-SULI compared to mCherry-SULI$^{KR}$, in which all the lysine residues in VVD were mutated to arginine (Supplementary Fig. 11). These results indicated that the SULI mediates protein degradation through a lysine ubiquitination-independent proteasome pathway. Therefore, SULI might be directly recognized and degraded by the proteasome in the dark state, which is similar to the case for the ornithine decarboxylase (ODC)[41] or cyclin-dependent kinase (Cdk) inhibitor p21Cip[42].

We then studied whether SULI could be used to control the degradation of target proteins inside subcellular organelles such as the nucleus or mitochondria by fusing localization signal sequences to the N-terminus of mCherry-SULI. The results showed that the dynamic range of light-induced stabilization of mCherry-SULI in the nucleus was 12-fold (Fig. 4d, Supplementary Fig. 12), slightly lower than that in the cytoplasm. However, minimal changes were observed for mCherry-SULI in mitochondria under light or dark conditions (Fig. 4d, Supplementary Fig. 12). It appears that SULI can efficiently mediate protein degradation in the cytoplasm and nucleus but not in mitochondria.

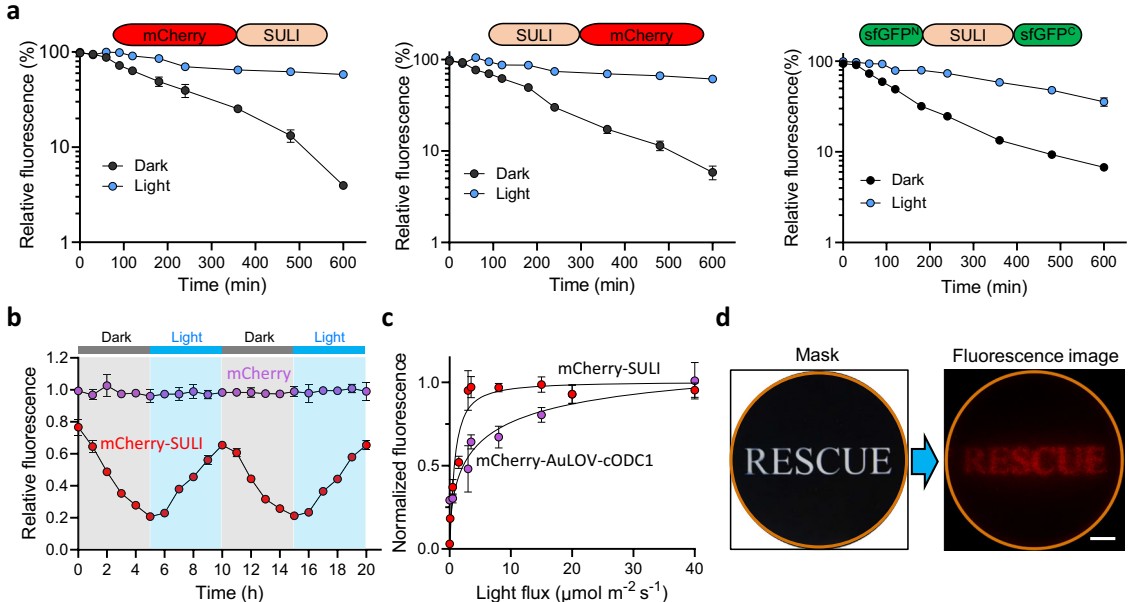

**Fig. 3 | Kinetics characteristics of SULI. a** Degradation kinetics of SULI-regulated protein stability. Yeast cells expressing different SULI fusion proteins cultured upon light illumination were then incubated with CHX and transferred to dark conditions or kept in light conditions. The fluorescence of the fusion proteins at the indicated time points was recorded by flow cytometry. The data were normalized to the fluorescence at time 0 min. **b** Oscillatory control of protein abundance by SULI. Yeast cells expressing mCherry-SULI were cultured alternatively in light and dark with an interval of 5 h for a total of 20 h. mCherry fluorescence at the indicated time points was measured using a microplate reader. The data were normalized to the data for the cells expressing mCherry alone at each indicated time point. The data

are presented as the mean ± SD from three biological replicates. **c** Light irradiance-dependent regulation of protein stability by SULI or AuLOV-cODC1. The data were collected by flow cytometry and normalized to the data for the cells expressing mCherry alone under each indicated blue light condition. The data in **a**–**c** are presented as the mean ± SD from three biological replicates. **d** Spatial control of protein abundance by SULI. Yeast cells expressing mCherry-SULI were grown on solid medium and irradiated by blue light using a mask with a specific image (left panel). Imaging of mCherry fluorescence was performed (right panel). The orange circle indicates the glass bottom of the plate to which the cells were attached. Scale bar, 1 cm. Source data are provided as a Source Data file.

These data are consistent with the knowledge that the proteasome pathways exist mainly in the cytosol and nucleus[43] and that mitochondrial matrix protein are degraded mainly by mitophagy or intrinsic proteases[44,45].

Intriguingly, the fluorescence microscopy data showed that mCherry-SULI aggregated before its degradation once the illuminated cells were transferred to dark conditions (Supplementary Fig. 13). In yeast, the Hsp104 protein functions as a disaggregase that cooperates with Ydj1p (Hsp40) and Ssa1p (Hsp70) to refold and reactivate denatured or aggregated proteins[46]. In an mCherry-SULI-expressing yeast strain containing GFP-tagged genomic Hsp104, good colocalization of mCherry and GFP fluorescence in the granules was observed when the light-illuminated cells were transferred to dark conditions until the mCherry fluorescence disappeared (Fig. 4e), suggesting that Hsp104 was recruited to the aggregated SULI fusion protein and mediated its degradation. In comparison, both the GFP and mCherry fluorescence exhibited diffuse distribution in yeast cells when they were kept under light conditions (Supplementary Fig. 14). In an Hsp104 knockout yeast strain simultaneously expressing sfGFP and mCherry-SULI, mCherry fluorescence aggregated when illuminated cells were transferred to dark conditions (Fig. 4f), while sfGFP was diffusely distributed; however, neither protein's fluorescence decreased significantly with time (Fig. 4f). Intriguingly, the aggregation of VVD-bearing fusions in the dark was also reported by Romano et al. However, the aggregates were quite stable and did not undergo degradation similar to SULI under dark conditions, probably due to different mutations in VVD and different growth temperatures and/or degradation pathways for yeast and E.coli cells[47]. Together, these observations suggest a model in which a conformational change in the SULI motif induces rapid aggregation of the SULI fusion protein, which is then bound by Hsp104 and other factors, disaggregated, refolded, and eventually recognized and degraded by the proteasome (Fig. 4g).

## Optical control of the cell cycle and growth by SULI

Cdc28, a cyclin-dependent kinase, is a major regulator of mitotic and meiotic cell cycles. Sic1, the substrate, and inhibitor of Cdc28, can control the G1/S phase transition through phosphorylation and degradation[42,43]. To test whether SULI can be used to control the yeast cell cycle and growth, we used SULI to tag the $^{\Delta N}$Sic1 protein, a shortened version of Sic1 that lacks the SCF$^{Cdc4}$-dependent degradation sequence within the N-terminal 105 amino acids and is thereby associated with a lengthened G1 phase (Fig. 5a). In the dark, the engineered yeast cells expressing $^{\Delta N}$Sic1-SULI grew similarly to the control cells (Fig. 5b). Upon light illumination, cell growth was significantly retarded, presumably due to accumulation of the $^{\Delta N}$Sic1-SULI protein (Fig. 5b). Further analysis revealed that the engineered yeast cells cultured under dark conditions had division behavior similar to that of the control cells (Fig. 5c, compare the inner rings in the mock $vs$ $^{\Delta N}$Sic1-SULI cells). Upon blue light irradiation, the majority of the $^{\Delta N}$Sic1-SULI cells displayed large buds with short spindles localized at the bud necks (Fig. 5c), the typical phenotype resulting from a blocked G1/S transition. The yeast cell cycle could also be manipulated by SULI through Clb2, another Cdc28 regulator that activates Cdc28 to promote the transition from G2 to M phase. As shown in Supplementary Fig. 15, fusion of SULI to the C-terminus or an inner position of a shortened variant of Clb2 lacking a destruction box motif (24–34 amino acids) for ubiquitin-mediated degradation by the proteasome[46] (Supplementary Fig. 15a, d) significantly prolonged metaphase (Supplementary Fig. 15b, c, e, f). Light-induced stabilization of $^{\Delta N}$Sic1 and Clb2 (Δde) was further validated by immunoblotting analysis, showing >10-fold On/Off switching ratios calculated from the signals of the bands (Supplementary Fig. 16).

We next tested whether SULI could control the stability of an endogenous protein. In a proof-of-principle study, we integrated the SULI-encoding sequence into the C-terminus of the genomic *ADE2*

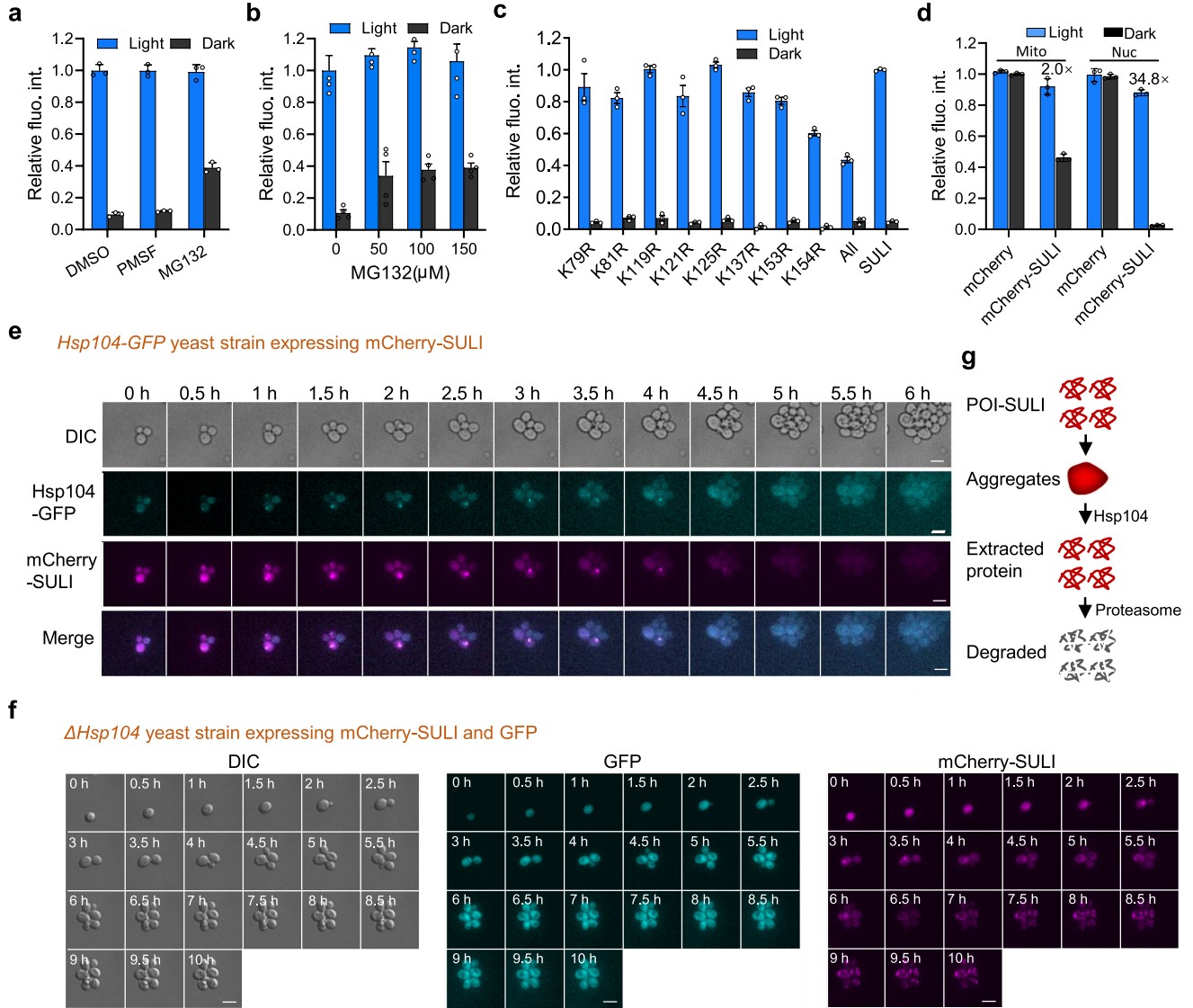

**Fig. 4 | Mechanism of SULI-mediated degradation. a** Effects of different inhibitors of degradation pathways on SULI-mediated degradation of target proteins. Overnight-cultured yeast cells expressing mCherry-SULI under light conditions were incubated with 75 μM MG132, 10 mM PMSF, or 10 mM DMSO for 6 h before mCherry fluorescence was determined by a microplate reader. **b** Effects of different concentrations of MG132. Overnight-cultured yeast cells expressing mCherry-SULI under light conditions were incubated with 0, 50, 100, or 150 μM MG132 for 6 h before mCherry fluorescence was determined a microplate reader. **c** Effects of SULI variants containing different mutations of Lys→Arg. "All" means that all Lys residues in SULI were mutated to Arg. Yeast cells expressing different fusions were cultured under light or dark conditions for 10 h before mCherry fluorescence was determined by flow cytometry. **d** SULI-regulated protein stability in the nucleus and mitochondria. Yeast cells expressing nucleus- or mitochondria-localized mCherry-SULI were cultured under light or dark conditions. The mCherry fluorescence of the cells was determined by flow cytometry. The data in **a–d** are presented as the mean ± SD from three biological replicates. **e, f** Role of the Hsp104 protein in SULI-regulated protein stability. Yeast strains expressing Hsp104-GFP **e** or with Hsp104 depletion **f** were used to express mCherry-SULI. The mCherry fluorescence of the cells transferred from light to dark conditions was imaged. Scale bar, 5 μm. **g** Proposed mechanism of SULI-mediated degradation. Source data are provided as a Source Data file.

gene, whose product is a phosphoribosyl aminoimidazole carboxylase that catalyzes an essential step in the 'de novo' purine nucleotide biosynthetic pathway (Fig. 5d). Our results revealed a light-dependent rescue of ADE2-SULI accumulation and adenine auxotrophy, which was not observed in control cells expressing ADE2-mCherry (Fig. 5e). Taken together, these results indicate that SULI can be used to optically control cellular activities through light-dependent regulation of protein stability.

## Optical control of protein stability by SULI in *zebrafish*

The zebrafish, a traditional model organism used to study development and diseases, has an optimal growth temperature similar to that of yeast, suggesting that SULI may be used to control protein stability in zebrafish. To test this possibility, we injected mRNA encoding mCherry-SULI together with mRNA encoding EGFP-Hsp104 or control mRNA encoding EGFP without fusion of Hsp104 into zebrafish embryos. The injected embryos were cultured in E3 medium supplemented with 5 μM FAD and kept under dark or light conditions for 24 h before mCherry fluorescence was determined (Fig. 6a). Our data showed that the intrinsic fluorescence of FAD had little interference with the GFP detection (Supplementary Fig. 17). When kept in the dark, embryos expressing mCherry-SULI showed nearly no detectable mCherry fluorescence, similar to the uninjected control embryos (Fig. 6b, c). Exposure to light led to marked red fluorescence in embryos expressing mCherry-SULI (Fig. 6b, c). Interestingly, the red fluorescence was significantly enhanced by the presence of Hsp104

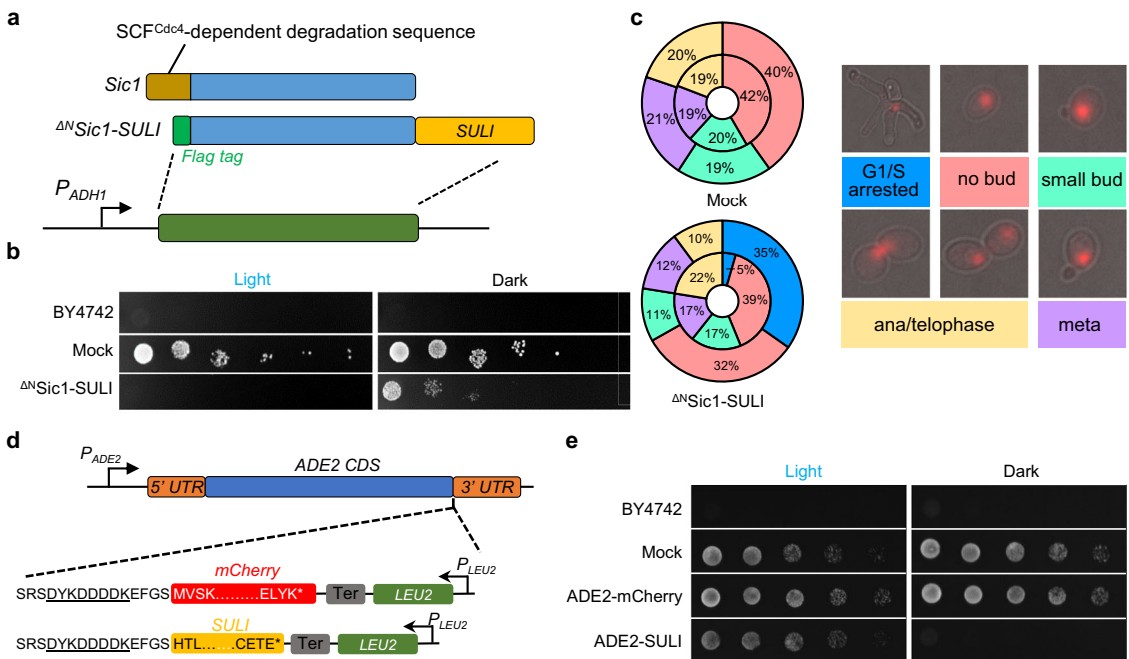

**Fig. 5 | Optical control of the cell cycle and growth. a** Schematic representation of the SULI-controlled stability of $^{\Delta N}$Sic1. SULI was fused to the C-terminus of $^{\Delta N}$Sic1, whose SCF$^{Cdc4}$-dependent degradation sequence (amino acids 1–105 of Sic1) was deleted. **b** Yeast cells expressing $^{\Delta N}$Sic1-SULI were serially diluted (1:10; first spot, ~2.5 × 10$^4$ cells) and grown in solid medium under light or dark conditions. BY4742 cells transformed with empty vectors were used as the controls. **c** The same yeast cells as in **b** were cultured under light or dark conditions for 10 h before imaging. The circular graphs (inner ring, dark conditions; outer ring, light conditions) show the mean distribution of cell cycle stages obtained from three biological replicates.

At least 100 cells were counted for each replicate. Propidium iodide was used to stain the DNA of yeast cells (red). Scale bars, 5 μm. **d** Schematic representation of SULI-mediated degradation of endogenous ADE2 expressed from the genomic gene. A DNA fragment containing the SULI-encoding gene and a LEU2 expression cassette was integrated into the C-terminus of the genomic *ADE2* gene. mCherry was used to replace SULI as the control. **e** Yeast cells expressing ADE2-SULI or ADE2-mCherry were serially diluted (1:5; first spot, ~2.5 × 10$^4$ cells) and grown in the solid medium under light or dark conditions.

(Fig. 6b, c). It seemed that Hsp104 was able to stabilize mCherry-SULI in embryos under light conditions, but the underlying mechanism remains to be resolved. In comparison, neither light nor Hsp104 altered the red fluorescence in the control embryos expressing mCherry only (Fig. 6b, c). These findings indicate that SULI is capable of regulating protein stability in zebrafish.

To further test the power of SULI as a genetic tool for zebrafish, we tagged Pitx2, a conserved homeodomain transcription factor that has multiple functions during embryonic development[48], with SULI. After injecting the mRNA encoding Pitx2-SULI and Hsp104 into zebrafish embryos, we found that 80% of the embryos kept in the dark had normal development, which was comparable to the percentage of embryos injected with the control mRNA and kept under the same conditions that had normal development (Fig. 6d–f). In contrast, only 42% of the Pitx2-SULI- and Hsp104-injected embryos developed normally when they were cultured under light illumination, while 58% died or were malformed with axial defects (Fig. 6d–f). Unfortunately, immunoblotting analysis of the light-induced stabilization of Pitx2 did not succeed. One of the potential reasons is that Pitx2 protein is unstable during embryonic development, leading to low level of protein accumulation in zebrafish. Collectively, our observations demonstrate that SULI is a useful tool with which to directly regulate protein function in zebrafish embryos.

## Discussion

Light, which enables manipulation at high temporal and spatial resolution, has been used to study diverse complex biological processes in live cells or in vivo using various optogenetics tools to control protein metabolism, activities, and interactions[49,50]. Among these tools, light-switchable control of gene expression or protein turnover is a general method that can be widely applied to different proteins. Light-inducible gene expression systems that we and others have developed control protein abundance at the transcriptional level[23–27]. Nevertheless, as mRNA transcription and protein translation take time, there is a significant delay between photoactivation and phenotype occurrence, which decreases the temporal resolution of the approach. Therefore, regulation of protein abundance via regulation of protein turnover may be more attractive than transcriptional regulation due to its simplicity and speed and hence is highly desirable. In the present study, we developed a highly sensitive and efficient tag termed SULI (stabilization upon light induction), which provides an alternative way to control the turnover of a protein in cells and in vivo.

The most prominent difference between SULI and previously reported light-switchable protein turnover tags (LIDs) is its operational mechanism. SULI is stabilized under light irradiance; in sharp contrast, LIDs show accelerated degradation under light irradiance. The opposite light responses of SULI and LIDs make them rather complementary, as each protein stability tag has differing characteristics and strengths. A protein tagged with SULI is maintained in an "OFF" or low-level state until light illumination. Conversely, for proteins tagged with LIDs, it is necessary to illuminate the cells with blue light continuously to maintain the "OFF" state. Therefore, SULI may be more convenient than LIDs in transgenic applications, as the cells or animals will need light irradiance only when induction is desired. SULI, in combination with LIDs, also provides an unprecedented opportunity to differentially control the turnover of multiple proteins, which may be particularly useful for the study of independent proteins with antagonistic functions. In addition, SULI can be combined with light-switchable transcription systems to reduce leaky expression in a uninduced state. In a proof-of-principle example, we fused SULI to an mCherry reporter driven by the LVAD light-switchable transactivator in yLightON system developed by our group[26]. Cells expressing mCherry-

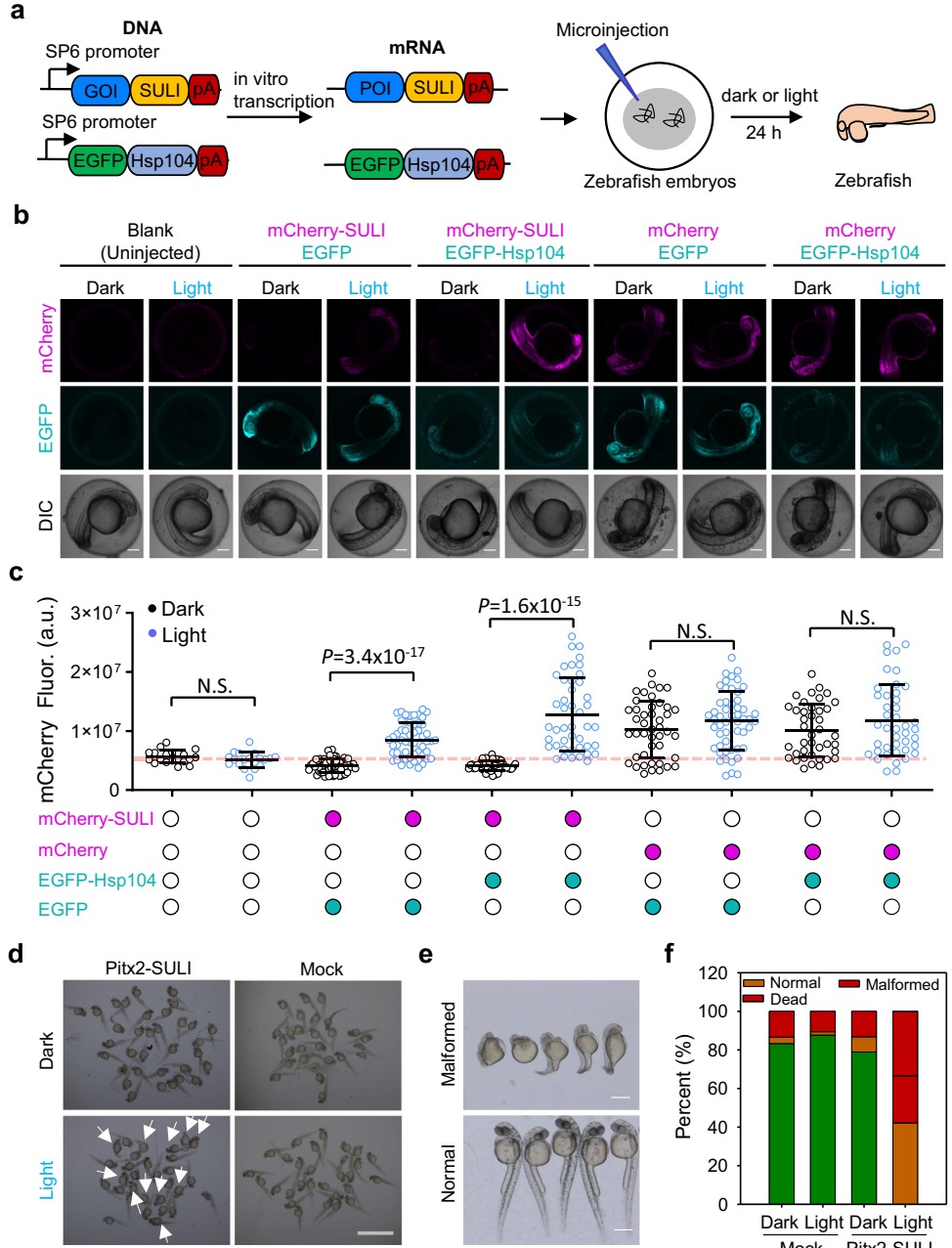

**Fig. 6 | Optical control of protein stability by SULI in *zebrafish*. a** Schematic representation of SULI-mediated degradation of the protein of interest (POI) in zebrafish embryos. mRNA encoding SULI fused with a protein of interest was transcribed in vitro using the SP6 promoter and injected into the zebrafish embryos. **b** Image of mCherry and EGFP fluorescence in zebrafish embryos injected with mRNAs encoding mCherry-SULI and EGFP or EGFP-Hsp104. mRNA encoding mCherry alone was used to replace mCherry-SULI as the control. All the zebrafish embryos were incubated in E3 medium supplemented with 5 µM FAD. Scale bar, 200 µm. **c** Quantitative analysis of mCherry fluorescence in zebrafish embryos. The data are presented as the mean ± SD. Statistical comparison was performed by two-tailed *t* test. *n* = 20, 19, 63, 56, 49, 44, 44, 52, 40, 42 zebrafish embryos from left to

right, respectively. N.S., no significant difference. a.u., arbitrary units. **d** Optical control of the development of zebrafish embryos by SULI. Zebrafish embryos injected with mRNA encoding Pitx2-SULI and Hsp104 were incubated under light or dark conditions for 24 hours before images were taken. mRNA encoding SULI and Hsp104 was used as the control. The white arrows indicate the malformed embryos. Scale bar, 200 µm. **e** Enlarged figures to show the malformation or normal phenotype of the injected zebrafish expressing Pitx2-SULI kept under light or dark conditions, respectively. Scale bar, 500 µm. **f** Quantitative analysis of the development of zebrafish embryos under different conditions. The data are the results from >100 embryos for each group. Source data are provided as a Source Data file.

SULI-controlled by LVAD exhibited significantly lower leakage under dark conditions, leading to a much higher On/Off ratio (>3000-fold), than cells expressing mCherry alone (Supplementary Fig. 18).

The simple and compact design of SULI, which consists of a single smallest LOV domain, offers additional benefits for protein turnover control. The small size of SULI and the ability of SULI to be placed at either a terminal or inner position of a target protein may minimize the

disruption of the fusion target's function, considering that many proteins need their N- or C-terminus to function and do not tolerate terminal fusions[51]. In comparison, LIDs have larger sizes and can only be fused at the C-terminus of a target protein. Furthermore, SULI does not require a degron sequence, thus reducing the possibility of undesired interaction with the host chassis. LIDs usually show low On/Off switching ratios and significant basal levels of degron activity that

destabilize the target protein in the dark[15,16,20]. For the most recently reported LID, psd3[17], the On/Off switching ratio was significantly improved to 40-fold after extensive engineering but at the expense of increased degradation of the ON state in the dark. In comparison, SULI showed a desirable 30-fold On/Off switching ratio, and it was rather stable in the ON state, with a half-life of over 10 hours. Such a difference in the degradation rate of the ON state of the stabilization tags was also reflected by their overall expression level, i.e., the expression of SULI under light conditions and of LIDs under dark conditions. Therefore, SULI may be more favorable than LIDs for applications when high levels of proteins are desired to exert their biological functions. Finally, SULI is a derivative of VVD, which has an unusually stable photoactivated state. The half-life of the light-activated state of VVD is $18,000 \, s$[28], unlike that of higher plant phototropins such as AtLOV and AsLOV, which were used to develop LIDs. We previously reported the LightOn gene expression system, which was engineered using a synthetic VVD module and is sensitive to light induction. Similarly, SULI also shows notable light sensitivity and minimal toxicity (Supplementary Fig. 19a), and even a pulsed illumination of low repetitive frequency (1 s of light and 29 s of dark) is sufficient for saturated activation of SULI (Supplementary Fig. 19b). Nevertheless, the light sensitivity of SULI limits its tunability and makes it difficult to fine-tune the protein levels in live yeast cells by adjusting the light irradiance. We also tested the ability of SULI in regulating protein stability in mammalian cells cultured at 37 °C. Unfortunately, we did not observe significant light-induced stabilization of mCherry reporter by SULI (Supplementary Fig. 20). Thus, SULI seems to be not adaptable to different organisms probably due to the distinct growth temperatures, protein degradation pathways, and/or cellular microenvironments for different organisms.

In summary, we developed SULI, a small and sensitive tag for stabilization upon light induction, that provides a powerful and convenient tool to study protein function and gene regulatory networks. SULI allows quantitative and spatiotemporal control of target protein in yeast cells, and can also be used for optogenetic control of protein in *zebrafish*. Thus, SULI may also be useful for regulating protein stability in other commonly used model organisms that grow at similar temperatures, such as *Drosophila* and *Caenorhabditis elegans*, although such applications were not studied in the current work. In addition, our previous studies have shown that introduction of mutations in VVD module could tune the photoinducible characteristics of VVD-based light-switchable factors, e.g, light sensitivity, decay kinetics, and switching dynamics, making these VVD-based optogenetic modules highly tunable[23,27,52]. In the future, SULI's properties, such as its degradation kinetics, dynamic range, and light sensitivity, may be fine-tuned by mutagenesis in the VVD domain following previously reported strategies for the development of light-switchable transcription factors and degrons.

## Methods

### Ethical statement
The zebrafish handling protocols were approved by the Ethical Review Committee of CAS Center for Excellence in Molecular Cell Science, Chinese Academy of Sciences (CAS), China.

### Yeast strains, growth conditions, and blue light irradiance
The *Saccharomyces cerevisiae* strains are derivatives of the S288C strain BY4742 (Genotype: MATα his3Δ1 leu2Δ0 lys2Δ0 ura3Δ0). Yeast strains used in this study are listed with genotype in Supplementary Table 2. The knock-in yeast strains with chromosomally integrated mCherry or SULI were obtained by homologous recombination with relevant PCR products directly. Briefly, primers containing the homologous sequences of genomic ADE2 expression cassette were used to amplify the DNA fragments encoding SULI gene and LEU2 expression cassette (See primer details in Supplementary Table 3). The obtained DNA fragments were integrated into the C-terminus of genomic ADE2 gene by standard chemical transformation. The clones were selected using solid medium lacking leucine under dark conditions and further verified colony PCR and Sanger sequencing.

For transformation, yeast cells were grown at 30 °C in YPD medium (1% yeast extract (Oxoid), 2% peptone (Oxoid), and 2% glucose (Sigma)) till OD = 0.6, and then transformed with plasmids or PCR products by the standard Li-Ac method. The transformed yeast cells were plated in synthetic drop-out solid media (TaKaRa). About 2 days later, the single yeast colony was picked, and subsequently confirmed by Sanger sequencing. Yeast cells for light-based patterning were grown on synthetic drop-out solid media (photomask covered; the photomask was created by printing a specific image onto the transparent film using a printer) with blue light at 26 °C for 2 days. If not mentioned particularly, the yeast cells were grown at 30 °C.

Unless otherwise indicated, for detection of light-regulated protein stability, cells were cultured in 48-well plates and illuminated by $15 \, \mu mol \, m^{-2} \, s^{-1}$ blue light emitting from an LED lamp (460 nm peak) or remained in the dark for 15 h before characterization. Neutral density filters were used to adjust the light intensity. Light intensities were measured with a luminometer (Sanwa, LX-2). For dark manipulation, a red (620–630 nm) LED lamp was used.

### Cloning
Unless stated otherwise, the DNA constructs were performed using the Hieff Clone One Step Cloning Kit (YEASEN). For expression in yeast, the pGADT7 AD vector (Clontech) was used as the basic backbone for the construction of the plasmids expressing different fusions. The SV40-NLS, GAL4 activation domain, and HA-tag were removed from the original pGADT7 AD vector by reverse PCR to obtain pGADH. The LEU2 nutritional marker was replaced with URA3 by reverse PCR and homologous recombination to obtain pGADH-U. Concretely, the DNA fragments of the vector and URA3 gene were generated by PCR, and then mixed with reagent (YEASEN) to produce the new vector. DNA fragments encoding multiple photosensitive proteins, including vivid, AsLOV2, ppLOV2, iLID, sLID and AtLOV, and mCherry were amplified from the synthetic constructs (Generay, Shanghai), and inserted into the NcoI and XhoI sites in the vector with the cloning kit. SULI mutants were generated by PCR approach with designed primers, and intramolecular recombination to get the mutant plasmids. The DNA fragments encoding the Sic1 and Clb2 were amplified from the yeast genome, and assembled with the DNA fragments of SULI, ADH1 promoter, and ADH1 terminator (amplified from pGADH) into the expression vector pRS313 to obtain the pRS313-Flag-ΔNSic1-SULI and pRS-Flag-ΔdbClb2-SULI. For expression in *E. coli*, the coding sequences of SUMO-SULI and SUMO-VVD were inserted into pET28a vector by traditional procedures or homologous recombination. The DNA fragments encoding SULI amplified from the pGADH expression vector and SUMO tag were inserted into pET28a vector with a 6xHis-tag at the N-terminus of the fusion proteins and a Strep-Tag II at the C-terminus of the fusion proteins by Gibson Assembly (Cat# E2611, NEB). For expression in zebrafish embryos, the DNA fragments encoding Hsp104 were amplified from the yeast genome, and assembled into the expression vector pTol2 with EGFP fragment by the Hieff Clone One Step Cloning Kit (YEASEN). The mCherry-SULI, mCherry, or Pitx2 (synthesized by Generay, Shanghai) fragment was also inserted into pTol2 with similar method to obtain the plasmids for zebrafish embryos. For expression in HEK 293 T cell line, the DNA fragments encoding mCherry-SULI and EGFP-Hsp104 were inserted into the pcDNA3.1 vectors with CMV promoter. All sequences of plasmids were confirmed by Sanger sequencing. The protein sequences of SULI and SULI_f were listed in Supplementary Note 2.

### Fluorescence measurement
Unless otherwise indicated, the overnight-cultured yeast cells were diluted (1:1000) into fresh medium and grown under light or dark

conditions for 10 hours. The fluorescence and $OD_{600}$ were measured using a Synergy 2 multimode microplate reader (BioTek) (ex = 590/20 nm and em = 645/40 nm for mCherry, ex = 485/20 nm and em = 528/20 nm for sfGFP). The fluorescence was normalized to the $OD_{600}$ of each sample. To analyze the fluorescence by flow cytometry, a CytoFLEX-S flow cytometer (Beckman Coulter) with an excitation of 561/10 nm and an emission of 610/20 nm for mCherry and an excitation of 488/8 nm and an emission of 525/40 nm for sfGFP was used. After acquisition, the raw cytometry data were processed and analyzed using the Cytexpert program (Beckman Coulter). The gating strategy for yeast cells was: (1) FSC-A/SSC-A gate for living cells; and 2) FSC-H/FSC-A and SSC-H/SSC-A for individual cells. To carry out the cycloheximide chase experiment, the engineered cells were cultured upon blue light illumination overnight and diluted to a density of $OD_{600}$ ~0.1. The cultures were then cultured under light or dark conditions to an $OD_{600}$ ~1.0 before 1 mM cycloheximide (CHX) was added to inhibit protein synthesis. The aliquots were taken at the indicated time points and mCherry fluorescence was analyzed using a Beckman CytoFLEX-S flow cytometry described above. For detection of mCherry fluorescence in mammalian cells, the transfected cells were harvested by trypsin digestion and resuspended with PBS buffer. The mCherry and EGFP fluorescence was analyzed using a Beckman CytoFLEX-S flow cytometer described above. The gating strategy for HEK293T cells was: (1) FSC-A/SSC-A gate for living cells; (2) FSC-H/FSC-A and SSC-H/SSC-A for individual cells; and (3) using the mock cells to define the gate for GFP positive cells.

## Western blot

The western blot were performed to analyze the protein level of the yeast cells containing SULI-relevant plasmids. The yeast cells were harvested by centrifugation (1800 × g, 2 min). The whole proteins of the cells were extracted by Yeast Buster Protein Extraction Reagent (Cat#71186, Millipore) and quantified by BCA Protein Quantification Kit (Cat# 20201ES76, Yeasen). 40 μg total proteins were resolved by 4−20% SDS−PAGE (Cat# M42015C, Genscript), and transferred to a PVDF membrane (Cat# ISEQ00010, Millipore). The membrane was blocked in 5% nonfat milk at room temperature for 1 h, followed by incubation with primary antibody on a shaker at 4 °C overnight. Primary antibodies listed below were used in this study: Monoclonal ANTI-FLAG M2 antibody (Cat# F1804, Sigma, 1:2000), Anti-mCherry antibody [1C51] (Cat# ab125096, Abcam, 1:2000) and HRP-conjugated GAPDH Mouse mAb [AMC0500] (Cat# AC035, Abclonal, 1:4000). The HRP-conjugated second antibodies (Anti-mouse IgG, HRP-linked Antibody, Cat#7076, Cell Signaling Technology, 1:4000) were detected using an enhanced chemiluminescence detection kit and visualized by ChemiDoc Touch Imaging System (Bio-Rad). The images were analyzed by ImageLab software (Bio-Rad). Uncropped western blots can be found in Source data.

## RNA purification and quantitative reverse transcription PCR

Total RNA was isolated from yeast cells with Spin Column Yeast Total RNA Purification Kit (Cat#B518657, Sangon Biotech, Shanghai) according to the manufacturer's instructions. cDNA was synthesized using PrimeScript RT reagent Kit with gDNA Eraser (Cat# RR047A, Takara). For quantitative reverse transcription PCR (RT-qPCR), 2 μl of the cDNA was used for the assay with 2× Universal SYBR Green Fast qPCR Mix (ABclonal) and specific primers on Roche LightCycle 480 Real-Time PCR System. The specificity of amplification was verified by melting-curve analysis, and the data were collected using the LightCycle 480 software. The procedures of qRT-PCR were set as following: initial denaturation, one cycle of 95 °C for 2 min; amplification, 40 cycles of 95 °C for 10 s and 60 °C for 30 s, with a final melting-curve analysis step (95 °C for 15 s and 60 °C for 60 s) to confirm the specificity of amplification and lack of primer dimers. All samples were normalized to the β-actin values by using the $2^{-\Delta\Delta Ct}$

formula. Primer sequences for PCR are listed in Supplementary Table 3.

## Protein purification and size-exclusion chromatography

pET28a-SUMO-SULI constructs were transformed into *Escherichia coli* BL21(DE3) cells. Single colonies were picked up into 500 ml LB media (50 μg/ml, Kanamycin). The cultured *E.coli* cells were induced at $OD_{600}$ = 0.6 with 1 mM IPTG and 10 μM FAD, and proteins were over-expressed for 24 h at 16 °C. The cells were harvested by centrifugation at 4 °C (1800 × g, 30 min). First, the proteins were extracted by ultrasonication. The soluble cell lysate was fractionated by centrifugation (10,000 × g, 30 min) and then purified with $Ni^{2+}$- affinity chromatography (Ni Sepharose, GE Healthcare). Elution buffer (100 mM HEPES, 150 mM NaCl, 300 mM imidazole, pH7.4) was used to elute the SUMO-SULI proteins. The purified proteins were diluted with equal volume buffer (100 mM HEPES, 150 mM NaCl, pH7.4), and then loaded into the Strep-Tactin XT 4Flow column (IBA Lifesciences). After washing by the Buffer W (Cat# 2-1003-100, IBA Lifesciences), Buffer BXT (Cat# 2-1042-025, IBA Lifesciences) containing 50 mM biotin was used to elute the bounded SUMO-SULI proteins. Then the SUMO-SULI proteins were desalted by Amicon Ultra centrifugal filter devices (Cat#UFC9003, Millipore) with the desalting buffer (12 mM HEPES, 135 mM KCl, 200 mM NaCl and 1 mM $MgCl_2$, pH7.7).

To assay the oligomerization states of the SUMO-SULI proteins, size-exclusion chromatography (SEC) was conducted with a size-exclusion chromatography column (Zenix SE-150, Sepax Technology). All protein samples were injected in 100 μL aliquots with a concentration range of 10 μM. The column was equilibrated with the mobile phase (12 mM HEPES, 135 mM KCl, 200 mM NaCl and 1 mM $MgCl_2$, pH7.7) at a flow rate of 1 mL/min. Light-state proteins were generated by blue light irradiation on ice for 5 min, and then immediately injected into the HPLC system.

## Imaging

Unless indicated, fluorescence imaging of yeast cells was performed using a Leica SP8 confocal laser scanning microscope equipped with an HC PL APO CS2 ×63.0/1.40 OIL objective and a HyD detector. mCherry fluorescence was imaged using a 561-nm laser and an emission wavelength range of 570−650 nm. sfGFP fluorescence was imaged using a 488-nm laser and an emission wavelength range of 500−550 nm. To detect the bioluminescence produced by NLuc, the engineered yeast cells expressing NLuc-SULI cultured under light or dark conditions were incubated with 25 μM furimazine (synthesized by ChemPartner and dissolved in 85% ethanol and 15% glycerol at a concentration of 5 mM as stock solution). The bioluminescence was imaged suing a Nikon Eclipse Ti2 microscope.

To detect the kinetics of Hsp104-mediated protein stability by SULI, the engineered cells were cultured until $OD_{600}$ reached ~1.0 under light conditions. The cells were then diluted to a density of $1 \times 10^3$ cells/μl using the fresh medium. One μl of the diluted cells was dropped onto the solid medium on surface of a glass slide and then covered the slide by a microscope coverslip. The cells were imaged using a Nikon Eclipse Ti-E microscope equipped with a Plan Apo ×40 DIC M N2 objective and a DS-Fi2 digital camera, using an FITC filter for GFP and a TxRed filter for mCherry.

## Cell cycle analysis for yeast cells

The plasmid encoding $^{\Delta N}Sic1$-SULI, $^{\Delta db}Clb2$-SULI or $Clb2(\Delta de)^N$-SULI-$Clb2(\Delta de)^C$ was transformed into BY4742 using a standard Li-Ac method. Above clones or clones containing genome-integrated ADE2-SULI or ADE2-mCherry expression cassette were picked and cultured until $OD_{600}$ reached approximately 0.6 under dark conditions. The cells were serially diluted (1:10; first spot was ~2.5 × 10⁴ cells) and cultured in solid medium under light or dark conditions for 3 days. The plates were imaged using a Tanon-5200.

For analysis of the detailed distribution of cell stages, the overnight-cultured cells under light or dark conditions were collected and fixed by incubation with prechilled 70% EtOH at 4 °C overnight. The cells were washed with 50 mM Na citrate buffer twice and resuspended with 50 mM Na citrate (containing 0.1 mg/ml RNase A). After 2 hours at 37 °C, an equal volume of 50 mM Na citrate (containing 8 µg/ml Propidium Iodide) were added and incubated for another 4 hours at 37 °C. Images of cell cycle stages were acquired using an Eclipse Ti inverted microscope system (Nikon) equipped with an S Plan Fluor ELWD ×40, 0.95 numerical aperture (NA) objective, and a digital sight camera.

## Cell culture and transfection
HEK293T (ATCC CRL-3216) (National Collection of Authenticated Cell Cultures, Center for Excellence in Molecular Cell Science, CAS) cells were maintained in Dulbecco's Modified Eagle's Medium with high glucose (HyClone) supplemented with 10% fetal bovine serum (Biological Industries) at 37 °C with 5% $CO_2$ and split every 2 days. Transfection of HEK293T cells was performed using the Hieff Trans Liposomal Transfection Reagent (Cat#40802ES02, YEASEN) according to the manufacturer's recommendations. Briefly, 0.4 µg of DNA and 1.2 µl of transfection reagent were mixed with 50 µl of Opti-MEM (GIBCO) and incubated for 20 min at room temperature. The mixture was added to each chamber of 35-mm four-chamber glass-bottom dishes with number 1 cover glass (Cellvis) with cells at 40–60% confluence.

## Zebrafish experiment
One-cell-stage zebrafish embryos (Danio rerio AB strain from CAS Center for Excellence in Molecular Cell Science, Chinese Academy of Sciences (CAS), China) were used in the zebrafish experiments. mCherry-SULI, EGFP-Hsp104, EGFP, Pitx2-SULI, and Hsp104 encoding genes were subcloned into a pTol2 vector, respectively, and then were used as the templates for amplifying the fragments containing SP6 promoter, coding sequences and SV40 poly(A) signal. The DNA fragments were purified and in vitro transcribed using a mMessage mMACHINE™ SP6 Transcription Kit (AM1340, Thermo) according to the manufacturer's manual. The DNA templates were digested by addition of two units of DNase, followed by mRNA precipitation using lithium chloride incubation for 2 hours at −20 °C.

For injections of the mRNA, 1 nL of the mRNA solution (150 ng/µl, mCherry-SULIs: EGFP /EGFP-Hsp104 = 4:1(w/w); 100 ng/µL, Pitx2-SULI, and Hsp104) was prepared and microinjected to the one-cell stage zebrafish embryo. The injected embryos were cultured at 28 °C in E3 medium (supplemented with 5 µM FAD) under blue light (5.4 µmol m$^{-2}$ s$^{-1}$) or dark conditions for 24 hours before fluorescence or morphology imaging was determined. Fluorescence images of the zebrafish larvae were taken using a Leica SP8 confocal laser scanning microscope equipped with a HCXPL APO 5.0 × 1.0 objective and a HyD detector, using an excitation of 488 nm and an emission of 500–550 nm for EGFP, an excitation of 561 nm and an emission of 570–650 nm for mCherry. Morphology of the zebrafish larvae was imaged using a zeiss stereomicroscope.

## Statistics and reproducibility
The data in most figure panels reflect multiple experiments performed using independent samples. Statistic differences between different groups were analyzed using the unpaired, two-tailed Student's $t$ test as indicated in the figure legend. GraphPad Prism and Microsoft Excel were used for plotting, data fitting, graphing, and statistical analysis. Statistical significance was determined as $P < 0.05$, with the exact $P$ values displayed in the figures. For experiments with replicates, the results were shown as mean ± s.d., unless stated otherwise. Representative western blot images were from at least three independent repeats with similar results. Representative micrographs were from at least three independent samples with similar results.

## Reporting summary
Further information on research design is available in the Nature Portfolio Reporting Summary linked to this article.

## Data availability
The data that support the findings of this study are available within the paper and supplementary information. The constructs generated in this study are available upon request from the corresponding authors with the appropriate Material Transfer Agreement (MTA). Source data are provided in this paper.

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

## Acknowledgements

We thank Dr. Zhengda Chen and Dr. Ni Su for imaging assistance and the other members of Yang lab for discussions and comments on this manuscript. We thank Dr. Jin-Qiu Zhou and Dr. Ming-Hong He (Shanghai Institute of Biochemistry and Cell Biology, CAS, China) for providing plasmids (pRS series) and discussions. We thank Professor Junbiao Dai (Center for Synthetic Genomics, Shenzhen Institutes of Advanced Technology, CAS) for providing yeast strains (BY4742, *ΔHsp104,* and *Hsp104-GFP* yeast strains). We thank Dr. Xiaoling Li (National Institute of Environmental Health Sciences, NIEHS) for her revision and suggestions on this manuscript. This research was supported by the National Key Research and Development Program of China (2022YFC3400100 to Y.Y and X.C.; 2019YFA0904800 to Y.Y.), NSFC (32121005, 32150028, 91857202 and 21937004 to Y.Y., 31971367 to M.M., 31971349 to X.C.), STI2030-Major Projects (2021ZD0202200 and 2021ZD0202203 to X.C.), the Shanghai Rising-Star Program to X.C., the Shanghai Municipal Education Commission-Frontier Science Research Base of Optogenetic Techniques for Cell Metabolism (2021 Sci & Tech 03-28) to Y.Y. and X.C., Research Program of State Key Laboratory of Bioreactor Engineering (to Y.Y. and X.C.), and the National Postdoctoral Program for Innovative Talents (BX20180336 to M.M.) and Shanghai Super Postdoctoral Program (to M.M.), the Fundamental Research Funds for the Central Universities to Y.Y. and X.C.

## Author contributions

Concepts were conceived by Y.Y., M.M., W.Z., and X.C.; M.M., Y.Q., W.Z., and S.Z. designed the experiments and analyzed the data; Z.W. gave technical support and conceptual advice. Y.Y., M.M., and X.C. wrote the manuscript.

## Competing interests

Y.Y., W.Z., and M.M. have submitted a patent application to the Chinese patent office pertaining to the development and characterization of SULI of this work (application number CN105177023B). The remaining authors declare no competing interests.
