## [Peer Review File · Nature Communications]

REVIEWER COMMENTS

Reviewer #1 (Remarks to the Author):

In this paper, Mao and colleagues present a new optogenetic tool for light-inducible stabilization of a protein of interest (POI): LIST. While several optogenetic tools exist that allow destabilizing a POI with light, currently only one tool was reported that is meant to stabilize a POI with blue light. This tool is based on the LOV domain of *Phaeodactylum tricornutum* aureochrome 1a (AuLOV) and effectively exploits the natural ability of this domain to become more stable under light. The difference in protein expression levels for dark and illuminated samples with AuLOV is, however, small (about 2-fold). The authors, therefore, rightly state that there is still the need for a robust, small tag for adjusting protein levels with light.

They started by noticing that the expression levels of fusion constructs containing the blue light photoreceptor VVD (a small LOV domain from *Neurospora crassa*) were higher in illuminated yeast cells than in those kept in the dark. We actually noticed the same in my lab when expressing VVD-bearing constructs in mammalian cells. They decided to build upon this interesting observation and to mutate VVD in an attempt to find variants with enhanced stability differences in the dark and under blue light. The authors find such mutants, and especially the triple mutant (Y50W, N56K or C71V) showed 30-fold difference in protein expression levels between light and dark conditions. The authors go on to characterize the tool using a cycloheximide chase experiment to prove that indeed the reason behind the expression level difference lies in the higher degradation rate of the fusion protein in the dark. They then test if dimerization of VVD is needed for the stabilization effect by implementing mutations known to inhibit dimerization. While the authors conclude that dimerization is not needed for the stabilization, I personally think the data presented are not sufficient to reach this conclusion (see points below). The authors nicely show that LIST can be fused on the POI at either terminus and even at internal positions, while the other tags can only be fused at the C-terminus of the POI. Using the proteasome inhibitor MG132, the authors show that the degradation of the fusion protein is due to the proteasome. They then create VVD mutants where each lysine was substituted with an arginine, but find no difference between the wt LIST and this mutant, suggesting that ubiquitylation be not the trigger in this case. The authors also show that LIST-mediated degradation of a POI works well for cytoplasmic and nuclear localization, but not mitochondrial localization. Fluorescence microscopy revealed that mCherry-LIST aggregates in yeast cells in the dark and that the chaperone Hsp104 plays a role in resolving the aggregates and facilitating LIST-bearing proteins' degradation. Finally, the authors apply LIST to control the cell cycle and growth in yeast and to control the function of the transcription factor Pitx2 in zebrafish.

I really like this paper. There is surely the need for a tool such as LIST in the cell biology community and I thank the authors for having gone to great lengths to move from the simple observation of the destabilizing effect that VVD has on any protein it is fused to when cells are kept in the dark to this robust and well-behaved tool, whose mechanism of action the authors at least partly characterized.

My major criticism to this nice work is that all data but those showed in Fig.5 and Suppl Fig. 12 are derived from *technical* replicates. Independent biological replicates are needed.

Here I mention my other major points (without a particular order), which I ask the authors to take into account:

1. Suppl Fig.4: most cells are not fluorescent. Only some cells are extremely bright. This seems to be happening in particular for the N56K C71V double mutant and the triple mutant. If this is only an impression coming from this specific image, the authors should show larger images and make it clear that all cells have fluorescence. FACS analysis would actually nicely give quantification and also show the variation in the population.
2. In the methods, I do read that flow cytometry was actually performed. However, I do not know which figures show data measured via flow cytometry rather than the plate reader. The authors should show histograms so that readers can see what is the variation of the expression levels within the population.
3. The Y40E mutant is said to have a higher On/Off ratio compared to the triple mutant. However, by eye, it seems to me that the slightly higher fluorescence level in the light is accompanied by a higher level in the dark, which would make this mutant less optimal than the one w/o the Y40E mutation. The authors should indicate the light/dark fold ratio in the figure.
4. This statement: "Further size-exclusion chromatography data showed that the variant with the Y40E mutation was a monomer in either the dark state or the light-activated state (Supplementary Fig. 7)" is not backed up by the data. The authors show only SEC under light conditions. They should also show SEC data for the protein in the dark in order to support a statement that dimerization is not important for the stabilization. It is important to convince the readers that the Y40E mutation inhibits dimerization.
5. Why were cells cultured for 24h for the experiment shown in Fig.2? In the other experiments (Suppl Figs up to Fig. S7) the cells were illuminated for 10 or 15 h.
6. Fig. S10: NLS-mCherry-LIST and NLS-mCherry show a very different localization pattern. NLS-mCherry appears to localize in smaller foci. Why is this? A nuclear marker in this experiment would help, since in yeast it is not that easy to identify the nucleus.
7. Aggregation of a construct carrying VVD as the photosensor has been recently reported by Romano and colleagues in *Escherichia coli* (Romano et al., *Nature Chemical Biology*, 2021). Interestingly, in *E. coli* the construct is stable. I think the authors should cite this paper and discuss the results, since in both cases aggregation of the VVD-bearing construct is observed in the dark.
8. Fig.6d: images are too small. I cannot recognize any malformation from the images. Please enlarge. If not possible in the main figure, add one figure in the supplement where the fish are clearly visible.

9. Fig. S13b: I do not understand what is γ LightOn-LIST. The authors have two columns, one is mCherry and one is mCherry-LIST. I thought they wanted to show the combination of γ LightOn with LIST, with LIST being fused to mCherry. Is LIST additionally fused to the GAVPO transcription factor? I am confused here. Additionally, the second column (indicated as γ LightOn) shows a very high mCherry level in the dark. This contradicts what has been published about the γ LightOn system (in the abstract of the paper "A Single-Component Optogenetic System Allows Stringent Switch of Gene Expression in Yeast Cells", it is written that the γ LightOn system has an *extremely low leakage*).

10. For the WB shown in Suppl Fig.S9: it is not clear to me if the GAPDH truly represents a loading control for the same membrane. First of all, Fig. S9 (and not S8 as written in the paper) does not show an "uncropped" WB for GAPDH. Second, from the method section, it seems the authors did not use fluorescence as detection method, but rather chemiluminescence. GAPDH would run at the same level as some of the other bands on the membrane, therefore I believe these are two different membranes, which means a loading control was not performed. This should be done and truly uncropped images should be shown.

11. Moreover, in the western blot shown in Suppl Fig.S9, the levels of mCherry are much higher than those of mCherry-LIST regardless of whether the sample was in the dark or illuminated. This strongly suggests that fusion to LIST dramatically decreases the expression level of the POI. This contradicts the data shown in Fig.2b. Yet, in Fig 3b, mCherry-LIST is 20% lower than just mCherry. Why is there this discrepancy across the data set?

12. Fig.6 is not very friendly to color-blind readers. The authors could replace red and green with other colors, which can be differentiated by all readers.

13. In the methods, the authors write cells were illuminated by 15 $\mu\text{mol m}^{-2} \text{s}^{-1}$ blue light emitting from an LED lamp (460 nm peak): does it mean that continuous illumination was used? Why not give pulses of blue light, considering that VVD remains in the illuminated state for 4-5 hours? Or does LIST have a shorter half-life? I also noticed that light was applied on the column during SEC and I asked myself the same question: why is this needed considering that the photo-adduct should last for much longer than the run?

14. Legend to Fig. S1: the authors write: "Yeast cells expressing different LOV domains fused fusions were cultured in light or dark conditions". Delete "fused".

15. It is a CHX chase experiment, not phase experiment.

16. In the introduction, the authors write: "Gene editing 1-4, transcriptional regulation 5, and RNA interference 6 are widely used practices". I would not call these practices, rather methods.

17. Line 12: I would write: chemically-induced

18. In the entire paper, the authors use N/C-terminal as the noun rather than the adjective. I would suggest to replace all these occurrences with C-terminus and N-terminus and leave N/C-terminal when it is the adjective.

19. I personally find this sentence confusing. "We further measured the degradation kinetics of VVD-regulated protein stability". I understand what the authors mean, yet speaking of degradation kinetics of

protein stability is odd. Perhaps it can be re-phrased as “We further measured the degradation kinetics of the VVD-bearing fusion protein” or something like this.

20. I would change this sentence: “LIST was able to be used to switch the abundance of the target protein” with this sentence: “LIST could be used to switch...”

21. In this sentence: “probably because it was difficult for MG132 to diffuse into the cells due to poor permeation of the yeast membrane, which was consistent with the previous studies 38” , I would delete *the* and write: “probably because it was difficult for MG132 to diffuse into the cells due to poor permeation of the yeast membrane, which was consistent with previous studies”.

22. I would change this sentence: “The yeast cell cycle was also able to be manipulated by LIST through Clb2” with this sentence: “The yeast cell cycle could also be manipulated by LIST...”

23. I would change this sentence: “As shown in Supplementary Fig. 12, fusion of LIST to the C-terminal or inner of a shortened variant of Clb2” with this sentence: ““As shown in Supplementary Fig. 12, fusion of LIST to the C-terminus or an inner position of a shortened variant of Clb2”

24. The authors write: “The zebrafish, a traditional model organism used to study development and diseases, has an optimal growth temperature similar to that of yeast, suggesting that LIST may be able to be used to control protein stability in zebrafish.” Does this mean that the authors think or know that the tool would not work at other temperatures? In the sentence, I suggest to replace “may be able to be used” with “may be used”

25. Exposure to light illumination: I would either write “Exposure to light” or “Illumination”

26. In this sentence: “It seemed that Hsp104 was able to stabilize mCherry-LIST in embryos under light conditions, but the underlying mechanism remained to be resolved”, I would write “remains to be resolved”

27. In this sentence: “To further test the probability of using LIST as a genetic tool for zebrafish”, I would rather write “To further test the power of LIST as a genetic tool for zebrafish”.

28. This sentence: “,considering that many proteins function with N- or C-terminal tags or both and do not tolerate terminal fusions 50.” is not clear because the tags in this case refer to functional motifs of the proteins and not tags such as LIST. I would rephrase it, for instance: “considering that many proteins need their N- or C-terminus to function and do not tolerate terminal fusions”

29. In the concluding sentence: “In the future, LIST’s properties, such as its degradation kinetics, dynamic range, and light sensitivity, may be able to be fine-tuned by mutagenesis in the VVD domain following previously reported strategies for the development of light-switchable transcription factors and degrons.” I would write “...and light sensitivity, may be fine-tuned by mutagenesis”

Reviewer #2 (Remarks to the Author):

This is a well written paper that describes an engineered variant of the protein VVD (called LIST) that can be fused to proteins of interest in order to regulate their abundance with light. LIST is relatively unique in that protein abundance is high under illumination and low in the dark. Most previously engineered photoswitches that regulate protein abundance work in the reverse direction, i.e. more protein in the dark. Another advantage of LIST is that proteins can be fused to either termini. The paper is appropriate for publication in Nature Communication after the following points are addressed.

- It should be stated explicitly in the abstract that the system has been tested in yeast and zebrafish (i.e. not mammalian cells). This is important as from the discussion in the manuscript it seems unlikely that the switch will be effective in mammalian systems.
- In the paper the authors mention that LIST is likely to work in systems that can be studied at lower temperatures (such as yeast and zebrafish). Reading between the lines, this implies that the switch does not work well at 37C. It seems like the authors must have tried the switch in mammalian cells. Does it not function properly at these conditions? If so, this would be valuable information to add to the paper, especially if there is data indicating why it is not working. Are there protein stability issues in the light and dark at 37C?
- In the paper, mCherry or sfGFP are used in all the experiments that quantitate protein levels with and without illumination. In the functional experiments performed with non-fluorescent proteins no explicit measurements are made of protein levels. Quantitative western blotting (or related techniques) should be used to show protein levels in these proteins before and after light illumination (Sic1, Clb2, Pitx2). From the functional assays it is clear that something is happening, but it would be good to know what fold-change in protein levels are leading to these effects. Are similar changes in proteins levels seen with these proteins as with the fluorescent proteins?
- On page 8, experiments are described with a mutant that changes the thermal reversion time of VVD. Please provide the reversion times for the mutant and WT VVD (can be from the literature).

Reviewer #3 (Remarks to the Author):

In this manuscript, Mao, Qian and Zhang et al. describe LIST as a new tool to optogenetically manipulate protein stability. The tool is based on the finding that the levels of the small LOV domain VVD are higher in the light than in the dark. Guided by previous knowledge about key residues regulating dimerization or thermal reversion of the protein, the authors introduced several mutations that increased the dynamic range of the tool. They show that LIST's functionality does not appear to depend on the position within a target protein, and they provide support for a ubiquitin-independent degradation pathway. While the relatively long half-life of 0.5-2 h precludes LIST's application in studies of highly dynamic processes, the authors demonstrate that it can be useful for observations of biological systems that are regulated on longer time scales in yeast and zebrafish. All experiments are very well executed and described, but the following points should be considered before publication to corroborate the findings and improve the presentation.

MAJOR POINTS

1) LIST characterization and degradation mechanism

a) In contrast to mCherry-AuLOV-cODC1, LIST shows almost no tunability. Please discuss this potential disadvantage and adjust the statements throughout the text.

b) The experiments with PMSF (and ideally also MG132 to potentially reveal the full inhibition range) need positive controls to be convincing. Please show that PMSF can penetrate cells and exert its effect in living yeast cells, or alternatively adjust the conclusions and statements.

c) Concerning line 692 and Fig. S9, ubiquitination has not been assessed directly with the chosen methods. The authors would need to show an anti-ubiquitin immunoblot after mCherry pulldown to be conclusive. Alternatively, please adjust the statements.

2) Zebrafish experiments

a) Based on ref. 48, where 50 pg mRNA were injected, strong axial defects are expected. Here, 100 pg Pitx2-LIST-encoding mRNA were injected, but strong axial defects are not apparent at the magnification and resolution shown in Fig. 6f. Please show larger magnifications and describe in more detail what "malformed" means in line 311. Please also discuss potential phenotypic discrepancies compared to ref. 48, or repeat the experiments if necessary.

b) Please calculate and mention in the manuscript how the light intensity for the zebrafish experiments mentioned in line 786 compares to the yeast experiments.

c) Does intrinsic fluorescence of FAD interfere with the zebrafish experiments? Did the medium for uninjected embryos also contain FAD? If intrinsic fluorescence is significant, please discuss this as a caveat in the manuscript.

d) The conclusion in line 371-372 is not valid, since some of the properties have only been shown in yeast but not in zebrafish. Therefore, please break up the sentence into two separate statements.

3) Data presentation

a) Please remove all preliminary data from the manuscript, so that only robust conclusions are published. Alternatively, please corroborate the findings to draw firm conclusions. This concerns the statements in lines 279-286 and 341-345.

b) Please show the raw data for Fig. 4d, i.e. not only the ratios. Please also show the condition in the light for Fig. 4e as a control (perhaps as a supplementary figure).

MINOR POINTS

1) Please change the conclusion in line 204 since LIST is not very rapid and no evidence of reversibility is presented.

2) Readers might perceive “light-inducible stabilization tag” as a misnomer, especially since tagging with LIST per se already seems to destabilize fusion partners such as mCherry. I therefore recommend to use an alternative name.

3) Please provide a rationale for the lysine-to-arginine mutations in lines 222-223 to guide the readers.

4) In the Reporting Summary, please make sure that all boxes are correctly checked in the section Flow Cytometry. For example, a figure exemplifying the gating strategy is not provided in the Supplementary Information etc. This also concerns the statements in the section “Code availability” of the Editorial Policy Checklist.

5) Please change cycloheximide “phase” to the more commonly used “pulse-chase” or “chase” (as written in the Materials and Methods section).

6) Please indicate directly in the legend of Fig.1 how long cells were grown in the dark and in the light.

7) Please explain the red stain in the legend of Fig. 5c (presumably propidium iodide).

8) Please explain how the mask mentioned in line 636 was created.

9) Please show the corresponding channels for mCherry in Fig. S5.

10) Is the data in Fig. S7 a spline or a polynomial fit to the raw data? If this is the case, please show the raw data.

11) I suggest to discuss possible applications in mammalian systems in lines 372-375 to broaden the impact.

12) Starting in line 304, the references seem to have gone out of register (e.g. ref. 45 in line 306 and ref. 48 in line 321 appear to be inappropriate).

We greatly thank the reviewers for the positive comments as well as the highly constructive suggestions to help improve our manuscript. We also sincerely apologize for the long delay for submitting our response. During the past year, my lab was closed several times caused by COVID-19 due to the following successive unfortunate events: 1) the environment of my lab was "false" positive in COVID-19 test caused by the contaminant of nucleic acid assay products of COVID-19 N gene (a student of mine once tried to develop rapid COVID-19 NA test); 2) COVID-19 caused citywide lockdown of Shanghai from March to June in 2022; 3) some students including one of my graduates became close contact of infected person. These events triggered immediate shut down of my facility several times with each long duration (2022.01, 2022.03-2022.06, 2022.09-2022.10, 2022.12), which significantly delayed our plan to revise the manuscript following the reviewers' suggestions. Now the country is less stringent COVID-19 control policy, and we hope this misfortune would not happen again in the future.

Reviewer #1 (Remarks to the Author):

In this paper, Mao and colleagues present a new optogenetic tool for light-inducible stabilization of a protein of interest (POI): LIST. While several optogenetic tools exist that allow destabilizing a POI with light, currently only one tool was reported that is meant to stabilize a POI with blue light. This tool is based on the LOV domain of *Phaeodactylum tricornutum* aureochrome 1a (AuLOV) and effectively exploits the natural ability of this domain to become more stable under light. The difference in protein expression levels for dark and illuminated samples with AuLOV is, however, small (about 2-fold). The authors, therefore, rightly state that there is still the need for a robust, small tag for adjusting protein levels with light.

They started by noticing that the expression levels of fusion constructs containing the blue light photoreceptor VVD (a small LOV domain from *Neurospora crassa*) were higher in illuminated yeast cells than in those kept in the dark. We actually noticed the same in my lab when expressing VVD-bearing constructs in mammalian cells. They decided to build upon this interesting observation and to mutate VVD in an attempt to find variants with enhanced stability differences in the dark and under blue light. The authors find such mutants, and especially the triple mutant (Y50W, N56K or C71V) showed 30-fold difference in protein expression levels between light and dark conditions. The authors go on to characterize the tool using a cycloheximide chase experiment to prove that indeed the reason behind the expression level difference lies in the higher degradation rate of the fusion protein in the dark. They then test if dimerization of VVD is needed for the stabilization effect by implementing mutations known to inhibit dimerization. While the authors conclude that dimerization is not needed for the stabilization, I personally think the data presented are not sufficient to reach this conclusion (see points below). The authors nicely show that LIST can be fused on the POI at either terminus and even at internal positions, while the other tags can only be fused at the C-terminus of the POI. Using the proteasome inhibitor MG132, the authors show that the degradation of the fusion protein is due to the proteasome. They then create VVD mutants where each lysine was substituted with an arginine, but find no

difference between the wt LIST and this mutant, suggesting that ubiquitylation be not the trigger in this case. The authors also show that LIST-mediated degradation of a POI works well for cytoplasmic and nuclear localization, but not mitochondrial localization. Fluorescence microscopy revealed that mCherry-LIST aggregates in yeast cells in the dark and that the chaperone Hsp104 plays a role in resolving the aggregates and facilitating LIST-bearing proteins' degradation. Finally, the authors apply LIST to control the cell cycle and growth in yeast and to control the function of the transcription factor Pitx2 in zebrafish.

I really like this paper. There is surely the need for a tool such as LIST in the cell biology community and I thank the authors for having gone to great lengths to move from the simple observation of the destabilizing effect that VVD has on any protein it is fused to when cells are kept in the dark to this robust and well-behaved tool, whose mechanism of action the authors at least partly characterized.

My major criticism to this nice work is that all data but those showed in Fig.5 and Suppl Fig. 12 are derived from *technical* replicates. Independent biological replicates are needed. Here I mention my other major points (without a particular order), which I ask the authors to take into account:

Response: We greatly thank the reviewer for the highly positive comments as well as the highly constructive suggestions to help improve our manuscript. We sincerely apologize for the confusion caused by the typo error. Indeed, almost all the data in the manuscript were obtained from at least three independent biological replicates. We have carefully checked all the data throughout the manuscript again and corrected the error to eliminate the confusion, and we greatly thank the reviewer again for pointing this out.

1. Suppl Fig.4: most cells are not fluorescent. Only some cells are extremely bright. This seems to be happening in particular for the N56K C71V double mutant and the triple mutant. If this is only an impression coming from this specific image, the authors should show larger images and make it clear that all cells have fluorescence. FACS analysis would actually nicely give quantification and also show the variation in the population.

Response: We thank the reviewer for pointing this out. As suggested by the reviewer, we have provided the FACS analysis of light-induced stabilization of mCherry reporter by different VVD mutants. The results showed that the engineered yeast cells expressing mCherry fused with different VVD mutants did exhibit markedly varied red fluorescence (**Supplementary Fig. 5b**). Similar phenomenon was also observed for the cells expressing mCherry alone (**Supplementary Fig. 5b**). Under dark conditions, the fluorescence of several mCherry-VVD variants significantly decreased and exhibited similar distributions with the mock yeast cells transformed with empty plasmid, whereas the cells expressing mCherry alone showed little difference in the fluorescence distributions under dark and light conditions (**Supplementary Fig. 5b**). These results demonstrate that the combinational mutations of VVD can be used for optogenetic control of protein stability.

We have integrated above results into the revised manuscript and greatly appreciate the reviewer's valuable suggestion.

2. In the methods, I do read that flow cytometry was actually performed. However, I do not know which figures show data measured via flow cytometry rather than the plate reader. The authors should show histograms so that readers can see what is the variation of the expression levels within the population.

Response: We thank the reviewer for the valuable suggestion. To avoid any confusion, we have indicated the methods of measurement in the figure legends, and we also have shown the histograms of some data obtained by flow cytometry to show the variation of the expression levels within the population following the reviewer's suggestion (**Supplementary Fig. 5, Supplementary Fig.10, and Supplementary Fig.12b**). Since substantial data in the manuscript were obtained by FACS, we did not show all the histograms of the data in current revision. If necessary, we are pleased to show them in the source data in the final revision of the manuscript.

3. The Y40E mutant is said to have a higher On/Off ratio compared to the triple mutant. However, by eye, it seems to me that the slightly higher fluorescence level in the light is accompanied by a higher level in the dark, which would make this mutant less optimal than the one w/o the Y40E mutation. The authors should indicate the light/dark fold ratio in the figure.

Response: We are sorry for the confusion. We have indicated the On/Off ratios of the mutants in **Fig. 1c** in the revised manuscript. The results showed that Y40E mutant had a higher fluorescence intensity under light conditions and a relatively lower fluorescence intensity under dark conditions compared to other mutants, which enabled the Y40E variant to have a highest On/Off ratio of >30-fold (**Fig. 1c**).

4. This statement: "Further size-exclusion chromatography data showed that the variant with the Y40E mutation was a monomer in either the dark state or the light-activated state (Supplementary Fig. 7)" is not backed up by the data. The authors show only SEC under light conditions. They show indeed show SEC data also for the protein in the dark in order to support a statement that dimerization is not important for the stabilization. It is important to convince the readers that the Y40E mutation inhibits dimerization.

Response: We thank the reviewer for the constructive suggestion. In the revised manuscript, we purified SUMO-LIST (SOMU tag is used to increase the soluble expression of LIST in E.coli) and measured its oligomeric states under light or dark conditions by a size-exclusion chromatography column using a high-performance liquid chromatograph (**Supplementary Fig. 8**). We also purified SUMO-VVD to be used as the controls. As shown by the data, the control SUMO-VVD was monomeric in the dark and underwent marked dimerization when illuminated with blue light, consistent with previous studies (Zoltowski et al., 2007). These data also confirmed that SUMO tag did not affect the oligomeric states of VVD (**Supplementary Fig. 8b**). In comparison, SUMO-LIST illuminated with blue light or kept in darkness shared the similar oligomeric state with the dark state (monomeric) of SOMU-VVD (**Supplementary Fig. 8b**), which supported the conclusion that the Y40E variant was monomeric in either the dark state or the light-activated state, consistent with previous studies (Vaidya et al., 2011). The small peaks after the main peak of the monomeric

state of SUMO-LIST might be the impure proteins shown in the SDS PAGE, which had little influence on the validation of the oligomeric state of SUMO-LIST.

We have integrated above results and description into the revised manuscript and greatly appreciate the reviewer's valuable suggestion.

5. Why were cells cultured for 24h for the experiment shown in Fig.2? In the other experiments (Suppl Figs up to Fig. S7) the cells were illuminated for 10 or 15 h.

Response: We thank the reviewer for catching this mistake. The culture time for the experiment shown in Fig.2 was 15 h. It is now corrected in the revised version.

6. Fig. S10: NLS-mCherry-LIST and NLS-mCherry show a very different localization pattern. NLS-mCherry appears to localize in smaller foci. Why is this? A nuclear marker in this experiment would help, since in yeast it is not that easy to identify the nucleus.

Response: We thank the reviewer for the constructive suggestion. Following the reviewer's suggestion, we used the DNA-intercalating fluorescent dye 4',6-diamidino-2-phenylindole (DAPI) for nucleus counterstaining of live yeast cells, which has been reported by previous literatures (Rinnerthaler et al., 2012; Zamostna et al., 2012). We did observe big blue fluorescent foci of nucleus in the cells, and majority of these foci co-localized well with the red fluorescent foci of NLS-mCherry or NLS-mCherry-LIST (**Supplementary Fig. 12a**). Notably, we also observed well co-localization of the smaller blue fluorescent foci with the mitochondria localized mCherry or mCherry-LIST (**Supplementary Fig.12a**), which was consistent with previous studies that DAPI can also be used for labeling of mitochondrial DNA (mtDNA) (Higuchi-Sanabria et al., 2016; Osman et al., 2015).

We have integrated above results into the revised manuscript and greatly appreciate the reviewer's valuable suggestion.

7. Aggregation of a construct carrying VVD as the photosensor has been recently reported by Romano and colleagues in Escherischia coli (Romano et al., Nature Chemical Biology, 2021). Interestingly, in E. coli the construct is stable. I think the authors should cite this paper and discuss the results, since in both cases aggregation of the VVD-bearing construct is observed in the dark.

Response: We thank the reviewer for the constructive suggestion. We have cited the literature and also added the following discussion in the revised manuscript following the reviewer's suggestion: *"Intriguingly, the aggregation of VVD-bearing fusions in the dark was also reported by Romano et al. However, the aggregates were quite stable and did not undergo degradation similar to LIST under dark conditions, probably due to different mutations in VVD and different growth temperatures and/or degradation pathways for yeast and E.coli cells."*(Line 251-255)

8. Fig.6d: images are too small. I cannot recognize any malformation from the images. Please enlarge. If not possible in the main figure, add one figure in the supplement where the fish are clearly visible.

Response: Following the reviewer's suggestion, we have added an enlarged figure to show the

malformation phenotype of the injected zebrafish expressing Pitx2-LIST and kept under light conditions (Fig. 6e). The results showed that many Pitx2-LIST injected embryos displayed severe spinal deformity under light conditions.

9. Fig. S13b: I do not understand what is γ LightOn-LIST. The authors have two columns, one is mCherry and one is mCherry-LIST. I thought they wanted to show the combination of γ LightOn with LIST, with LIST being fused to mCherry. Is LIST additionally fused to the GAVPO transcription factor? I am confused here. Additionally, the second column (indicated as γ LightOn) shows a very high mCherry level in the dark. This contradicts what has been published about the γ LightOn system (in the abstract of the paper “A Single-Component Optogenetic System Allows Stringent Switch of Gene Expression in Yeast Cells”, it is written that the γ LightOn system has an *extremely low leakage*).

Response: We sincerely apologize for the confusion. In fact, we combined LIST with the original eLightOn system (before optimization) in which LAVD instead of LAVDO was used as the light-switchable transcription factor. The data in Xu et al. showed that LVAD had a relatively high leak expression under dark conditions even though a weak promoter was used to drive its expression to reduce the light-independent dimerization, leading to a low On/Off ratio (23-fold) (Figure 1C in Xu et al.). After extensive optimization of the VVD variants, VVD mutant with the triple mutations Y50W/I85V/M135I (optimized LVAD (LVADO)) showed a high On/Off ratio of 573-fold. In this study, we combined LIST and LVAD by fusing LIST to an mCherry reporter driven by the LVAD light-switchable transcription factor. The results showed that the combined system had a dramatically decreased leaky expression compared to LVAD alone under the uninduced state, resulting in a much higher On/Off ratio of >3,000-fold.

We have revised Fig. S13 (new **Supplementary Fig.18**) and the corresponding description to eliminate the confusion, and greatly appreciate the reviewer’s valuable suggestion.

10. For the WB shown in Suppl Fig.S9: it is not clear to me if the GAPDH truly represents a loading control for the same membrane. First of all, Fig. S9 (and not S8 as written in the paper) does not show an “uncropped” WB for GAPDH. Second, from the method section, it seems the authors did not use fluorescence as detection method, but rather chemiluminescence. GAPDH would run at the same level as some of the other bands on the membrane, therefore I believe these are two different membranes, which means a loading control was not performed. This should be done and truly uncropped images should be shown.

Response: We have shown the original WB image for GAPDH in **Supplementary Note 1**. For the WB experiment, we first visualized the bands of mCherry and its fusion proteins (left panel). Thereafter, the membrane was stripped and reprobbed to visualize GAPDH. In the right panel of GAPDH signal, we still could see the residual signals of mCherry and its fusion proteins that have not been stripped cleanly, which strongly supported the fact that the GAPDH came from the same membrane with mCherry and its fusion proteins.

To eliminate the confusion, we have provided the uncropped images in the revised manuscript (**Supplementary Note 1**) and greatly appreciate the reviewer’s valuable suggestion.

11. Moreover, in the western blot shown in Suppl Fig.S9, the levels of mCherry are much higher than those of mCherry-LIST regardless of whether the sample was in the dark or illuminated. This strongly suggests that fusion to LIST dramatically decreases the expression level of the POI. This contradicts the data shown in Fig.2b. Yet, in Fig 3b, mCherry-LIST is 20% lower than just mCherry. Why is there this discrepancy across the data set?

Response: The reviewer raised the concern that why there was discrepancy across the data set regarding to the relative expression levels of mCherry and mCherry-LIST. It should be aware that the data produced with western blot is typically considered to be semi-quantitative, as there are many factors that can affect the signals of target protein, e.g., antibody affinity, transfer efficiency, protein configuration, linear range and internal loading control (McDonough et al., 2015; Pillai-Kastoori et al., 2020). Thus, it is not recommended to precisely quantify the levels of target proteins just according to the signals of the bands, especially when the target proteins have different sequences, molecular weights and configurations. In our experiment, we intended to use WB to validate whether the target proteins had distinct stabilities under light and dark conditions, rather than to accurately compare the expression levels between mCherry and mCherry-LIST. The difference in the signals of mCherry and mCherry-LIST bands could not reflect the real difference in their expression levels, as their molecule weights and protein configurations were different, which might affect the membrane transferring and the antibody binding in the WB experiments. In comparison, determination of the fluorescence intensity is a more straightforward way to precisely quantify the levels of a target protein fused with a fluorescent protein (such as GFP or mCherry), which has been widely used in biological studies (He et al., 2021; Lo et al., 2015). Therefore, we compared the levels of mCherry and mCherry-LIST by measuring their fluorescence intensity directly in our studies.

We have integrated above discussion into the revised manuscript and greatly appreciate the reviewer's valuable suggestion (Line 1013-1021).

12. Fig.6 is not very friendly to color-blind readers. The authors could replace red and green with other colors, which can be differentiated by all readers.

Response: We have modified the colors in Fig. 6b and c in the revised manuscript and greatly appreciate the reviewer's valuable suggestion.

13. In the methods, the authors write cells were illuminated by $15 \mu\text{mol m}^{-2} \text{s}^{-1}$ blue light emitting from an LED lamp (460 nm peak): does it mean that continuous illumination was used? Why not give pulses of blue light, considering that VVD remains in the illuminated state for 4-5 hours? Or does LIST have a shorter half-life? I also noticed that light was applied on the column during SEC and I asked myself the same question: why is this needed considering that the photo-adduct should last for much longer than the run?

Response: We strongly agree with the reviewer's opinion that a suitable light irradiance is crucial for an optogenetic system. We did use $15 \mu\text{mol m}^{-2} \text{s}^{-1}$ continuous light illumination for optogenetic control of protein stability in our experiments. Our data showed that such light

illumination had little effect on the growth of yeast cells (**Supplementary Fig. 19a**). In the revised manuscript, we systematically studied the effects of different pulses of blue light on the light-induced stabilization by LIST following the reviewer's suggestion. As shown by the data, the pulsed illumination strategy could also be applied to effectively activate LIST, and even a pulsed illumination of low repetitive frequency (1 s of light and 29 s of dark) was sufficient for saturated activation of LIST (**Supplementary Fig. 19b**).

In our first submission, we studied the oligomeric states of LIST by a size-exclusion chromatography column using a AKTA system, which typically took more than one hour for the entire procedure. To keep the light state during the chromatography process, we applied continuous light illumination during the running process. However, it seemed that continuous light illumination was not necessary according to above results. In the revised manuscript, we used a high-performance liquid chromatograph (HPLC) system, which typically took 10 minutes to complete the analysis (**Supplementary Fig. 8**). Thus, we did not illuminate the column during the analysis again.

We have integrated above results and description into the revised manuscript and greatly appreciate the reviewer's valuable suggestion.

14. Legend to Fig. S1: the authors write: "Yeast cells expressing different LOV domains fused fusions were cultured in light or dark conditions". Delete "fused".

Response: We have deleted the "fused" in the revised manuscript following the reviewer's suggestion.

15. It is a CHX chase experiment, not phase experiment.

Response: We have revised the description following the reviewer's suggestion.

16. In the introduction, the authors write: "Gene editing 1-4, transcriptional regulation 5, and RNA interference 6 are widely used practices". I would not call these practices, rather methods.

Response: We have revised the description following the reviewer's suggestion.

17. Line 12: I would write: chemically-induced

Response: We have revised the description following the reviewer's suggestion.

18. In the entire paper, the authors use N/C-terminal as the noun rather than the adjective. I would suggest to replace all these occurrences with C-terminus and N-terminus and leave N/C-terminal when it is the adjective.

Response: We have revised the description following the reviewer's suggestion.

19. I personally find this sentence confusing. "We further measured the degradation kinetics of VVD-regulated protein stability". I understand what the authors mean, yet speaking of degradation kinetics of protein stability is odd. Perhaps it can be re-phrased as "We further measured the degradation kinetics of the VVD-bearing fusion protein" or something like this.

Response: We have re-phrased the sentence following the reviewer's suggestion.

20. I would change this sentence: "LIST was able to be used to switch the abundance of the target protein" with this sentence: "LIST could be used to switch..."

Response: We have re-phrased the sentence following the reviewer's suggestion.

21. In this sentence: "probably because it was difficult for MG132 to diffuse into the cells due to poor permeation of the yeast membrane, which was consistent with the previous studies 38", I would delete *the* and write: "probably because it was difficult for MG132 to diffuse into the cells due to poor permeation of the yeast membrane, which was consistent with previous studies".

Response: We have re-phrased the sentence following the reviewer's suggestion.

22. I would change this sentence: "The yeast cell cycle was also able to be manipulated by LIST through Clb2" with this sentence: "The yeast cell cycle could also be manipulated by LIST..."

Response: We have re-phrased the sentence following the reviewer's suggestion.

23. I would change this sentence: "As shown in Supplementary Fig. 12, fusion of LIST to the C-terminal or inner of a shortened variant of Clb2" with this sentence: "As shown in Supplementary Fig. 12, fusion of LIST to the C-terminus or an inner position of a shortened variant of Clb2"

Response: We have re-phrased the sentence following the reviewer's suggestion.

24. The authors write: "The zebrafish, a traditional model organism used to study development and diseases, has an optimal growth temperature similar to that of yeast, suggesting that LIST may be able to be used to control protein stability in zebrafish." Does this mean that the authors think or know that the tool would not work at other temperatures? In the sentence, I suggest to replace "may be able to be used" with "may be used"

Response: In the revised manuscript, we tested the ability of LIST in regulating protein stability in mammalian cells cultured at 37°C. Unfortunately, we did not observe significant light-induced stabilization of mCherry reporter by LIST (**Supplementary Fig. 20**). Thus, temperature seems to have a significant effect on the performance of LIST in regulating protein stability in different organisms, which may be related to the temperature-sensitive photoinducible characteristics of VVD reported previously (Hunt et al., 2007).

We have integrated above results into the revised manuscript and greatly appreciate the reviewer's valuable suggestion. We also have re-phrased the sentence following the reviewer's suggestion.

25. Exposure to light illumination: I would either write "Exposure to light" or "Illumination"

Response: We have revised the description following the reviewer's suggestion.

26. In this sentence: "It seemed that Hsp104 was able to stabilize mCherry-LIST in embryos under light conditions, but the underlying mechanism remained to be resolved", I would write "remains

to be resolved”

Response: We have re-phrased the sentence following the reviewer’s suggestion.

27. In this sentence: “To further test the probability of using LIST as a genetic tool for zebrafish”, I would rather write “To further test the power of LIST as a genetic tool for zebrafish”.

Response: We have re-phrased the sentence following the reviewer’s suggestion.

28. This sentence: “,considering that many proteins function with N- or C-terminal tags or both and do not tolerate terminal fusions 50.” is not clear because the tags in this case refer to functional motifs of the proteins and not tags such as LIST. I would rephrase it, for instance: “considering that many proteins need their N- or C-terminus to function and do not tolerate terminal fusions”

Response: We have re-phrased the sentence following the reviewer’s suggestion.

29. In the concluding sentence: “In the future, LIST’s properties, such as its degradation kinetics, dynamic range, and light sensitivity, may be able to be fine-tuned by mutagenesis in the VVD domain following previously reported strategies for the development of light-switchable transcription factors and degrons.” I would write “...and light sensitivity, may be fine-tuned by mutagenesis”

Response: We have re-phrased the sentence following the reviewer’s suggestion.

Reviewer #2 (Remarks to the Author):

This is a well written paper that describes an engineered variant of the protein VVD (called LIST) that can be fused to proteins of interest in order to regulate their abundance with light. LIST is relatively unique in that protein abundance is high under illumination and low in the dark. Most previously engineered photoswitches that regulate protein abundance work in the reverse direction, i.e. more protein in the dark. Another advantage of LIST is that proteins can be fused to either termini. The paper is appropriate for publication in Nature Communication after the following points are addressed.

Response: We greatly thank the reviewer for the highly positive comments as well as the highly constructive suggestions to help increase the quality of our manuscript.

- It should be stated explicitly in the abstract that the system has been tested in yeast and zebrafish (i.e. not mammalian cells). This is important as from the discussion in the manuscript it seems unlikely that the switch will be effective in mammalian systems.

Response: We have revised the abstract following the reviewer's suggestion.

- In the paper the authors mention that LIST is likely to work in systems that can be studied at lower temperatures (such as yeast and zebrafish). Reading between the lines, this implies that the switch does not work well at 37C. It seems like the authors must have tried the switch in mammalian cells. Does it not function properly at these conditions? If so, this would be valuable information to add to the paper, especially if there is data indicating why it is not working. Are there protein stability issues in the light and dark at 37C?

Response: As suggested by the reviewer, we tested the performance of LIST in regulating protein stability in mammalian cells cultured at 37 °C. Unfortunately, we did not observe significant light-induced stabilization of mCherry reporter by LIST (**Supplementary Fig. 20**). Thus, temperature seems to have significant effect on the performance of LIST in regulating protein stability in different organisms, which may be related to the temperature-sensitive photoinducible characteristics of VVD reported previously (Hunt et al., 2007).

We have integrated above results and discussion in the revised manuscript and greatly appreciate the reviewer's valuable suggestion.

- In the paper, mCherry or sfGFP are used in all the experiments that quantitate protein levels with and without illumination. In the functional experiments performed with non-fluorescent proteins no explicit measurements are made of protein levels. Quantitative western blotting (or related techniques) should be used to show protein levels in these proteins before and after light illumination (Sic1, Clb2, Pitx2). From the functional assays it is clear that something is happening, but it would be good to know what fold-change in protein levels are leading to these effects. Are similar changes in proteins levels seen with these proteins as with the fluorescent proteins?

Response: We are very thankful to the reviewer for his/her suggestions to validate the protein levels before and after light illumination in the functional assays, which definitely strengthened

the quality of our manuscript. As he/she suggested we performed western blotting analysis of Δ^N Sic1-LIST and Clb2(Δ de)-LIST levels in yeast cells cultured under light or dark conditions. We observed significant light-induced stabilization for both Δ^N Sic1-LIST and Clb2(Δ de)-LIST, the On/Off switching ratios of stabilization could reach 26.4- and 10.5-fold, respectively (**Supplementary Fig. 16, Supplementary Note 1**), which were slightly lower than that of mCherry-LIST. Notably, Δ^N Sic1-LIST appeared to have a molecule weight between 70 kDa and 100 kDa, which was significantly larger than 40.1 kDa according to its amino acid sequence. It seemed that the shifted band was the phosphorylated form of Δ^N Sic1-LIST, as the endogenous Sic1 needs to be phosphorylated at multiple sites for ubiquitination-driven degradation (Moreno-Torres et al., 2017; Nash et al., 2001), and the phosphorylated form of a protein is typically significantly heavier than its nonphosphorylated counterpart (Moreno-Torres et al., 2017; Venta et al., 2020).

We also attempted to validate the status of Pitx2-LIST in zebrafish cultured under light or dark conditions using western blotting. Unfortunately, we observed no specific band even though systematical optimization of the WB process was performed, including the primary antibody (anti-Pitx2, anti-flag or anti-HA), the chemiluminescence detection kit, transfer conditions and so on. Alternatively, we also used a fluorescence method to validate the status of Pitx2-LIST by tagging Pitx2-LIST with GFP. The obtained Pitx2-GFP-LIST fusion allowed optogenetic control of embryonic development similar to Pitx2-LIST, but no significant green fluorescence was observed in zebrafish under both light and dark conditions. Notably, we did not find immunoblot analysis of Pitx2 expression in zebrafish studies according to the literatures either. Thus, it seems that Pitx2 has an extremely low expression during the embryonic development of zebrafish, making it very difficult to be detected.

We have integrated above results and description into the revised manuscript and greatly appreciate the reviewer's valuable suggestion.

- On page 8, experiments are described with a mutant that changes the thermal reversion time of VVD. Please provide the reversion times for the mutant and WT VVD (can be from the literature).

Response: As suggested by the reviewer, we have provided the reversion times for the WT VVD and its mutants from the literature in **Supplementary Table 1**.

Reviewer #3 (Remarks to the Author):

In this manuscript, Mao, Qian and Zhang et al. describe LIST as a new tool to optogenetically manipulate protein stability. The tool is based on the finding that the levels of the small LOV domain VVD are higher in the light than in the dark. Guided by previous knowledge about key residues regulating dimerization or thermal reversion of the protein, the authors introduced several mutations that increased the dynamic range of the tool. They show that LIST's functionality does not appear to depend on the position within a target protein, and they provide support for a ubiquitin-independent degradation pathway. While the relatively long half-life of 0.5-2 h precludes LIST's application in studies of highly dynamic processes, the authors demonstrate that it can be useful for observations of biological systems that are regulated on longer time scales in yeast and zebrafish. All experiments are very well executed and described, but the following points should be considered before publication to corroborate the findings and improve the presentation.

Response: We appreciate the reviewer's overall positive evaluation, specifically, that all our experiments are very well executed and described.

MAJOR POINTS

1) LIST characterization and degradation mechanism

a) In contrast to mCherry-AuLOV-cODC1, LIST shows almost no tunability. Please discuss this potential disadvantage and adjust the statements throughout the text.

Response: We are not absolutely sure about the reviewer's point here, since we did not demonstrate the tunability of LIST in the manuscript. However, our previous studies have shown that introduction of mutations in VVD module could tune the photoinducible characteristics of VVD-based light-switchable factors, e.g, light sensitivity, decay kinetics, and switching dynamics, making these VVD-based optogenetic modules highly tunable (Chen et al., 2016; Li et al., 2020; Liu et al., 2022). Thus, LIST is also likely to have high tunability by modulating the photoinducible characteristics of VVD in the future.

We have integrated above discussion into the revised manuscript following the reviewer's suggestion (Line 389-393).

b) The experiments with PMSF (and ideally also MG132 to potentially reveal the full inhibition range) need positive controls to be convincing. Please show that PMSF can penetrate cells and exert its effect in living yeast cells, or alternatively adjust the conclusions and statements.

Response: The reviewer raised the concern that whether PMSF can penetrate cells and exert its effect in living yeast cells. In fact, PMSF has been used in numerous living yeast studies (Kametaka et al., 1996; Lee and Goldberg, 1996; Takeshige et al., 1992), in particular the mechanisms of autophagy by Ohsumi group (2016 Nobel Prize for Physiology or Medicine). For example, Lee et al. carried out an exhaustive study to compare the effects of different inhibitors in different proteolytic pathways, which well demonstrated the penetration of PMSF into live yeast cells to distinguish the cytosolic and vacuolar proteolytic pathways (Lee and Goldberg, 1996).

Furthermore, we used the ERG6 mutant strain to increase the permeability of small chemicals in our studies (Graham et al., 1993; Nitiss and Wang, 1988), which further facilitated PMSF to enter the cells .

We have integrated part of above description into the revised manuscript and greatly appreciate the reviewer's suggestion (Line 215-216).

c) Concerning line 692 and Fig. S9, ubiquitination has not been assessed directly with the chosen methods. The authors would need to show an anti-ubiquitin immunoblot after mCherry pulldown to be conclusive. Alternatively, please adjust the statements.

Response: Our fluorescence and immunoblot analysis had shown that mCherry-LIST^{KR} (all eight lysine residues in VVD were mutated to arginine) exhibited similar light-induced stabilization with mCherry-LIST (**Fig. 4c and Supplementary Fig.11**), demonstrating that LIST-mediated degradation is ubiquitin independent, since there are no lysine sites in LIST^{KR} that can be ubiquitinated. To make the description more rigorous, we have adjusted the statements in the revised manuscript and greatly appreciate the reviewer's valuable suggestion (Line 739).

2) Zebrafish experiments

a) Based on ref. 48, where 50 pg mRNA were injected, strong axial defects are expected. Here, 100 pg Pitx2-LIST-encoding mRNA were injected, but strong axial defects are not apparent at the magnification and resolution shown in Fig. 6f. Please show larger magnifications and describe in more detail what "malformed" means in line 311. Please also discuss potential phenotypic discrepancies compared to ref. 48, or repeat the experiments if necessary.

Response: The reviewer raised the concern that why injection of more mRNA did not lead to more apparent phenotype. As shown by the data (**Fig. 1 and 2**), fusion with LIST could lead to a certain decrease in the expression level of the target protein. Similarly, the expression level of 100 pg Pitx2-LIST mRNA was probably less than that of the equal amount of Pitx2 mRNA. Therefore, it was difficult to directly compare the phenotypes of 100 pg Pitx2-LIST and 50 pg Pitx2 mRNA, as their expression levels were not linear.

We also have provided the larger magnifications of the malformed zebrafish in new **Fig. 6e**. As shown by the figure, many Pitx2-LIST injected embryos displayed severe spinal deformity under light conditions, consistent with the phenotypes shown in the literature (Ji et al., 2016).

We have integrated above results into the revised manuscript and greatly appreciate the reviewer's valuable suggestion.

b) Please calculate and mention in the manuscript how the light intensity for the zebrafish experiments mentioned in line 786 compares to the yeast experiments.

Response: We have indicated the light intensity for the zebrafish experiments in the revised manuscript following the reviewer's suggestion.

c) Does intrinsic fluorescence of FAD interfere with the zebrafish experiments? Did the medium for uninjected embryos also contain FAD? If intrinsic fluorescence is significant, please discuss this

as a caveat in the manuscript.

Response: The reviewer's concern should be whether there was potential interference of the intrinsic fluorescence of FAD with GFP detection, since they shared similar fluorescence spectra (ex: 365-465 nm, em: 520-530 nm for FAD (Galban et al., 2016); ex: 488 nm, em: 507 nm for GFP (Shaner et al., 2005)). We used the same medium containing FAD for both the uninjected and injected embryos in our previous experiment. To test it, we detected the fluorescence of embryos incubated in medium with or without FAD at the GFP channel. We did not observe significant fluorescence for the embryos incubated in medium containing 5 μ M FAD compared to those incubated in medium containing no FAD (**Supplementary Fig. 17**), indicating that the intrinsic fluorescence of FAD had minimal interference with the GFP detection.

We have integrated above results into the revised manuscript and greatly appreciate the reviewer's valuable suggestion (Line 298-299).

d) The conclusion in line 371-372 is not valid, since some of the properties have only been shown in yeast but not in zebrafish. Therefore, please break up the sentence into two separate statements.

Response: We have broken up the sentence into two separate statements following the reviewer's suggestion (Line 383-385).

3) Data presentation

a) Please remove all preliminary data from the manuscript, so that only robust conclusions are published. Alternatively, please corroborate the findings to draw firm conclusions. This concerns the statements in lines 279-286 and 341-345.

Response: We sincerely apologize for the confusion caused by the inappropriate phrase. All the data in the manuscript have been carried out independently more than two times with similar results. To eliminate the confusion, we have revised the description and also added the statement "at least two independent experiments were carried out with similar results" in the corresponding figure legend (Line 639).

b) Please show the raw data for Fig. 4d, i.e. not only the ratios. Please also show the condition in the light for Fig. 4e as a control (perhaps as a supplementary figure).

Response: We have provided the raw data for Fig. 4d (new **Fig. 4d**), and we also have showed the condition in the light for Fig. 4e as a control in the revised manuscript following the reviewer's suggestion (**Supplementary Fig. 14**).

MINOR POINTS

1) Please change the conclusion in line 204 since LIST is not very rapid and no evidence of reversibility is presented.

Response: We have revised the description following the reviewer's suggestion.

2) Readers might perceive "light-inducible stabilization tag" as a misnomer, especially since tagging

with LIST per se already seems to destabilize fusion partners such as mCherry. I therefore recommend to use an alternative name.

Response: Compared to the unstable properties of LIST in the dark, we prefer to focus the light-inducible stabilization characteristics of LIST, so we decide to use LIST to name the light-inducible stabilization tag, but still greatly appreciate the reviewer's suggestion.

3) Please provide a rationale for the lysine-to-arginine mutations in lines 222-223 to guide the readers.

Response: We have provide a rationale for the lysine-to-arginine mutations following the reviewer's suggestion.

4) In the Reporting Summary, please make sure that all boxes are correctly checked in the section Flow Cytometry. For example, a figure exemplifying the gating strategy is not provided in the Supplementary Information etc. This also concerns the statements in the section "Code availability" of the Editorial Policy Checklist.

Response: We thank the reviewer for pointing this out, and we have re-checked the Reporting Summary and Editorial Policy Checklist.

5) Please change cycloheximide "phase" to the more commonly used "pulse-chase" or "chase" (as written in the Materials and Methods section).

Response: We have revised the description following the reviewer's suggestion.

6) Please indicate directly in the legend of Fig.1 how long cells were grown in the dark and in the light.

Response: We have provided the information in the figure legend following the reviewer's suggestion.

7) Please explain the red stain in the legend of Fig. 5c (presumably propidium iodide).

Response: We thank the reviewer for pointing this out. The red stain was propidium iodide, and we have added the description in the figure legend.

8) Please explain how the mask mentioned in line 636 was created.

Response: The photomask was created by printing a specific image onto the transparent film using a printer. We have added the description in the revised manuscript.

9) Please show the corresponding channels for mCherry in Fig. S5.

Response: We are sorry for the confusion, and no mCherry signal was detected in Fig. S5. We have corrected the typo error in the figure legend.

10) Is the data in Fig. S7 a spline or a polynomial fit to the raw data? If this is the case, please show the raw data.

Response: Fig. S7 in our original submission was the raw data obtained from the GE AKTA. In the revised manuscript, we have modified Fig. S7 (new **Fig. S8**) with new figure following reviewer 1's suggestion. In the new Fig. S8, the curves were created from the raw data obtained by a HPLC analysis, and we did not perform extra fit for these data.

11) I suggest to discuss possible applications in mammalian systems in lines 372-375 to broaden the impact.

Response: We have added the discussion following the reviewer's suggestion.

12) Starting in line 304, the references seem to have gone out of register (e.g. ref. 45 in line 306 and ref. 48 in line 321 appear to be inappropriate).

Response: We have revised the references following the reviewer's suggestion.

List of new or updated figures

Fig. 1c, 3c, 4d, 6b, 6c and 6e

Supplementary Fig.5, 8, 10, 12, 14, 16, 17, 18, 19 and 20

References

Chen, X., Liu, R., Ma, Z., Xu, X., Zhang, H., Xu, J., Ouyang, Q., and Yang, Y. (2016). An extraordinary stringent and sensitive light-switchable gene expression system for bacterial cells. *Cell research* 26, 854-857.

Galban, J., Sanz-Vicente, I., Navarro, J., and de Marcos, S. (2016). The intrinsic fluorescence of FAD and its application in analytical chemistry: a review. *Methods Appl Fluoresc* 4, 042005.

Graham, T.R., Scott, P.A., and Emr, S.D. (1993). Brefeldin A reversibly blocks early but not late protein transport steps in the yeast secretory pathway. *The EMBO journal* 12, 869-877.

He, L., Tan, P., Zhu, L., Huang, K., Nguyen, N.T., Wang, R., Guo, L., Li, L., Yang, Y., Huang, Z., et al. (2021). Circularly permuted LOV2 as a modular photoswitch for optogenetic engineering. *Nature chemical biology* 17, 915-923.

Higuchi-Sanabria, R., Garcia, E.J., Tomoiaga, D., Munteanu, E.L., Feinstein, P., and Pon, L.A. (2016). Characterization of Fluorescent Proteins for Three- and Four-Color Live-Cell Imaging in *S. cerevisiae*. *PLoS one* 11, e0146120.

Hunt, S.M., Elvin, M., Crosthwaite, S.K., and Heintzen, C. (2007). The PAS/LOV protein VIVID controls temperature compensation of circadian clock phase and development in *Neurospora crassa*. *Genes & development* 21, 1964-1974.

Ji, Y., Buel, S.M., and Amack, J.D. (2016). Mutations in zebrafish *pitx2* model congenital malformations in Axenfeld-Rieger syndrome but do not disrupt left-right placement of visceral organs. *Developmental biology* 416, 69-81.

Kametaka, S., Matsuura, A., Wada, Y., and Ohsumi, Y. (1996). Structural and functional analyses of APG5, a gene involved in autophagy in yeast. *Gene* 178, 139-143.

Lee, D.H., and Goldberg, A.L. (1996). Selective inhibitors of the proteasome-dependent and vacuolar pathways of protein degradation in *Saccharomyces cerevisiae*. *J Biol Chem* *271*, 27280-27284.

Li, X., Zhang, C., Xu, X., Miao, J., Yao, J., Liu, R., Zhao, Y., Chen, X., and Yang, Y. (2020). A single-component light sensor system allows highly tunable and direct activation of gene expression in bacterial cells. *Nucleic acids research* *48*, e33.

Liu, R., Yang, J., Yao, J., Zhao, Z., He, W., Su, N., Zhang, Z., Zhang, C., Zhang, Z., Cai, H., et al. (2022). Optogenetic control of RNA function and metabolism using engineered light-switchable RNA-binding proteins. *Nature biotechnology* *40*, 779-786.

Lo, C.A., Kays, I., Emran, F., Lin, T.J., Cvetkovska, V., and Chen, B.E. (2015). Quantification of Protein Levels in Single Living Cells. *Cell reports* *13*, 2634-2644.

McDonough, A.A., Veiras, L.C., Minas, J.N., and Ralph, D.L. (2015). Considerations when quantitating protein abundance by immunoblot. *American journal of physiology. Cell physiology* *308*, C426-433.

Moreno-Torres, M., Jaquenoud, M., Peli-Gulli, M.P., Nicastro, R., and De Virgilio, C. (2017). TORC1 coordinates the conversion of Sic1 from a target to an inhibitor of cyclin-CDK-Cks1. *Cell discovery* *3*, 17012.

Nash, P., Tang, X., Orlicky, S., Chen, Q., Gertler, F.B., Mendenhall, M.D., Sicheri, F., Pawson, T., and Tyers, M. (2001). Multisite phosphorylation of a CDK inhibitor sets a threshold for the onset of DNA replication. *Nature* *414*, 514-521.

Nitiss, J., and Wang, J.C. (1988). DNA topoisomerase-targeting antitumor drugs can be studied in yeast. *Proceedings of the National Academy of Sciences of the United States of America* *85*, 7501-7505.

Osman, C., Noriega, T.R., Okreglak, V., Fung, J.C., and Walter, P. (2015). Integrity of the yeast mitochondrial genome, but not its distribution and inheritance, relies on mitochondrial fission and fusion. *Proceedings of the National Academy of Sciences of the United States of America* *112*, E947-956.

Pillai-Kastoori, L., Schutz-Geschwender, A.R., and Harford, J.A. (2020). A systematic approach to quantitative Western blot analysis. *Analytical biochemistry* *593*, 113608.

Rinnerthaler, M., Buttner, S., Laun, P., Heeren, G., Felder, T.K., Klinger, H., Weinberger, M., Stolze, K., Grousl, T., Hasek, J., et al. (2012). Yno1p/Aim14p, a NADPH-oxidase ortholog, controls extramitochondrial reactive oxygen species generation, apoptosis, and actin cable formation in yeast. *Proceedings of the National Academy of Sciences of the United States of America* *109*, 8658-8663.

Shaner, N.C., Steinbach, P.A., and Tsien, R.Y. (2005). A guide to choosing fluorescent proteins. *Nature methods* *2*, 905-909.

Takehige, K., Baba, M., Tsuboi, S., Noda, T., and Ohsumi, Y. (1992). Autophagy in yeast demonstrated with proteinase-deficient mutants and conditions for its induction. *The Journal of cell biology* *119*, 301-311.

Vaidya, A.T., Chen, C.H., Dunlap, J.C., Loros, J.J., and Crane, B.R. (2011). Structure of a light-activated LOV protein dimer that regulates transcription. *Science signaling* *4*, ra50.

Venta, R., Valk, E., Ord, M., Kosik, O., Paabo, K., Maljavin, A., Kivi, R., Faustova, I., Shtaida, N., Lepiku, M., et al. (2020). A processive phosphorylation circuit with multiple kinase inputs and mutually diversional routes controls G1/S decision. *Nature communications* *11*, 1836.

Zamostna, B., Novak, J., Vopalensky, V., Masek, T., Burysek, L., and Pospisek, M. (2012). N-terminal domain of nuclear IL-1alpha shows structural similarity to the C-terminal domain of Snf1 and binds to the HAT/core module of the SAGA complex. *PloS one* *7*, e41801.

Zoltowski, B.D., Schwerdtfeger, C., Widom, J., Loros, J.J., Bilwes, A.M., Dunlap, J.C., and Crane, B.R. (2007). Conformational switching in the fungal light sensor Vivid. *Science* *316*, 1054-1057.

REVIEWERS' COMMENTS

Reviewer #1 (Remarks to the Author):

I was truly wondering what had happened to this nice paper, since I was indeed expecting to receive the revised version sooner than this. I am very sorry for the authors for the misfortunate events that took place in between. Good to know that things went back to normal.

The authors did everything I asked for. I think it is time for the community to read it.

It is a pity that LIST does not work in mammalian cells..I was so much looking forward to using it...!

One last remark: in the SEC, I see there are impurities in the purification, however it is interesting that the tail is seen only in the illuminated sample. There seems to be something going on with the LIST construct itself. It is however surely not dimerisation.

Congratulations to the authors for this work.

Reviewer #2 (Remarks to the Author):

My concerns have been addressed with this revision. The paper is appropriate for publication in Nature Communications.

Reviewer #3 (Remarks to the Author):

The authors have appropriately addressed most of the reviewers' comments, but I'm still concerned regarding the following conclusions:

1) In their revised manuscript the authors state that "LIST allows rapid, reversible, quantitative, and spatiotemporal control of protein turnover". I had previously pointed out that LIST is not very rapid and

no evidence of reversibility is presented, but the authors have not changed their conclusions in the revised manuscript.

2) As I had pointed out previously, the role of ubiquitination has not been addressed, and the conclusions should be adjusted. The authors believe that they can rule out ubiquitination with their lysine mutants, but it is well known that ubiquitination can also occur on other non-lysine amino acid residues.

3) In my previous review, I had pointed out that in contrast to mCherry-AuLOV-cODC1, LIST shows almost no tunability. This referred to the data presented in the current Fig. 3c, which shows that LIST acts in an on/off-like manner whereas mCherry-AuLOV-cODC1 levels are tunable over a range of light intensities. In their revised manuscript, the authors have not discussed this potential disadvantage.

4) The authors now describe that they observed zebrafish embryos “with severe spinal deformity (Fig. 6d-f)”. As I had pointed out previously, it would be more appropriate to refer to the phenotypes as “axial defects” if they indeed reflect the observations in ref. 48.

5) In the revised manuscript, the authors now state that “Unfortunately, immunoblotting analysis of the light-induced stabilization of Pitx2 did not succeed, probably due to the extremely low expression of Pitx2 during the embryonic development of zebrafish”. However, this statement cannot be made in the absence of any experimental evidence. Actually, injection of 100 pg of mRNA – as was done in the current manuscript – is at the upper limit of typical zebrafish injections, which can often be as low as 1-10 pg and where visualization of the protein by direct fluorescence, immunoblot or immunostainings is possible.

6) In the rebuttal, the authors discuss classical papers that would provide easy readouts for the positive controls that I had requested (e.g. autophagic bodies visualized by simple microscopy from the experiments shown in Fig. 4a). Such positive controls would bolster the implied conclusion that vacuolar-lysosomal pathways are not involved in LIST-mediated degradation.

7) The new conclusion "Thus, temperature seems to have a significant effect on the performance of LIST in regulating protein stability in different organisms" does not directly follow from the data and should be adjusted.

8) The new conclusion "It is also possible for LIST to optogenetically control protein stability in mammalian cells after further systematical optimization of the photoinducible characteristics of VVD." also does not directly follow from the data and should be adjusted.

9) Similar to my previous assessment, reviewer 1 stated that "Moreover, in the western blot shown in Suppl Fig.S9, the levels of mCherry are much higher than those of mCherry-LIST regardless of whether the sample was in the dark or illuminated. This strongly suggests that fusion to LIST dramatically decreases the expression level of the POI". As I had pointed out previously, readers might therefore perceive "light-inducible stabilization tag" as a misnomer since tagging with LIST per se already seems to destabilize fusion partners such as mCherry. However, the authors decided not to use an alternative name.

10) I do not agree with the new phrasing that "It should be aware that the data produced with western blot is typically considered to be semi-quantitative, as there are many factors that can affect the signals of target protein, e.g., antibody affinity, transfer efficiency, protein configuration, linear range and internal loading control 1, 2. Thus, it is not recommended to precisely quantify the levels of target proteins just according to the signals of the bands, especially when the target proteins have different sequences, molecular weights, and configurations. Instead, determination of the fluorescence intensity is a more straightforward way to precisely quantify the levels of a target protein fused with a fluorescent protein (such as GFP or mCherry), which has been widely used in biological studies 3, 4.". The authors point to the correct guidelines for quantitative western blotting, and relevant experimental data following these guidelines can indeed be found in the literature. On the other hand, it is not correct that "determination of the fluorescence intensity is a more straightforward way to precisely quantify the levels of a target protein fused with a fluorescent protein". In fact, fluorescence intensities can be prone to structural changes; particularly prominent examples are nanobodies that can change fluorescence intensity without affecting protein abundance – here, using fluorescence intensity as a proxy for protein levels would be misleading.

11) The paper now lists a patent application and deleted "and materials" from the "Data and materials availability" section. The authors should include a statement that the constructs are freely available without any restrictions.

REVIEWERS' COMMENTS

Reviewer #1 (Remarks to the Author):

I was truly wondering what had happened to this nice paper, since I was indeed expecting to receive the revised version sooner than this. I am very sorry for the authors for the unfortunate events that took place in between. Good to know that things went back to normal. The authors did everything I asked for. I think it is time for the community to read it. It is a pity that LIST does not work in mammalian cells. I was so much looking forward to using it...! One last remark: in the SEC, I see there are impurities in the purification, however it is interesting that the tail is seen only in the illuminated sample. There seems to be something going on with the LIST construct itself. It is however surely not dimerization. Congratulations to the authors for this work.

Response: We greatly thank the reviewer for the positive comments as well as the highly constructive suggestions to help increase the quality of our manuscript. We greatly appreciate the reviewer's careful examination of the data, and we prefer to investigate the interesting phenomenon in an independent study.

Reviewer #2 (Remarks to the Author):

My concerns have been addressed with this revision. The paper is appropriate for publication in Nature Communications.

Response: We greatly thank the reviewer for the positive comments as well as the highly constructive suggestions to help increase the quality of our manuscript.

Reviewer #3 (Remarks to the Author):

The authors have appropriately addressed most of the reviewers' comments, but I'm still concerned regarding the following conclusions:

Response: We greatly thank the reviewer for the positive comments as well as the highly constructive suggestions to help increase the rigor of our conclusions in the manuscript.

1) In their revised manuscript the authors state that "LIST allows rapid, reversible, quantitative, and spatiotemporal control of protein turnover". I had previously pointed out that LIST is not very rapid and no evidence of reversibility is presented, but the authors have not changed their conclusions in the revised manuscript.

Response: As suggested by the reviewer, we have revised all the statements throughout the manuscript.

2) As I had pointed out previously, the role of ubiquitination has not been addressed, and the conclusions should be adjusted. The authors believe that they can rule out ubiquitination with their lysine mutants, but it is well known that ubiquitination can also occur on other non-lysine amino acid residues.

Response: To make the conclusion more rigorous, we have revised the conclusion as “*SUL1 mediates protein degradation through a lysine ubiquitination-independent proteasome pathway*” following the reviewer’s suggestion .

3) In my previous review, I had pointed out that in contrast to mCherry-AuLOV-cODC1, LIST shows almost no tunability. This referred to the data presented in the current Fig. 3c, which shows that LIST acts in an on/off-like manner whereas mCherry-AuLOV-cODC1 levels are tunable over a range of light intensities. In their revised manuscript, the authors have not discussed this potential disadvantage.

Response: As suggested by the reviewer, we have discussed the potential disadvantage of SUL1 tag in the revised manuscript (Lines 375-377).

4) The authors now describe that they observed zebrafish embryos “with severe spinal deformity (Fig. 6d-f)”. As I had pointed out previously, it would be more appropriate to refer to the phenotypes as “axial defects” if they indeed reflect the observations in ref. 48.

Response: As suggested by the reviewer, we have corrected the description in the revised manuscript.

5) In the revised manuscript, the authors now state that “Unfortunately, immunoblotting analysis of the light-induced stabilization of Pitx2 did not succeed, probably due to the extremely low expression of Pitx2 during the embryonic development of zebrafish”. However, this statement cannot be made in the absence of any experimental evidence. Actually, injection of 100 pg of mRNA – as was done in the current manuscript – is at the upper limit of typical zebrafish injections, which can often be as low as 1-10 pg and where visualization of the protein by direct fluorescence, immunoblot or immunostainings is possible.

Response: To eliminate the confusion, we have revised the description as “*Unfortunately, immunoblotting analysis of the light-induced stabilization of Pitx2 did not succeed. One of the potential reasons is that Pitx2 protein is unstable during the embryonic development, leading to low level of protein accumulation in zebrafish*”, and greatly appreciate the reviewer’s constructive suggestion (Lines 316-317).

6) In the rebuttal, the authors discuss classical papers that would provide easy readouts for the positive controls that I had requested (e.g. autophagic bodies visualized by simple

microscopy from the experiments shown in Fig. 4a). Such positive controls would bolster the implied conclusion that vacuolar-lysosomal pathways are not involved in LIST-mediated degradation.

Response: We agree with the reviewer's opinion that further validation of the status of yeast vacuole upon PMSF treatment would bolster the implied conclusion that vacuolar-lysosomal pathways are not involved in LIST-mediated degradation. To make the conclusion more rigorous, we have revised the conclusion as "*In comparison, PMSF may have better membrane permeability and has been extensively used in numerous living yeast studies, but further detection of the status of vacuole upon PMSF treatment will be helpful to bolster the conclusion that vacuolar-lysosomal pathways are not involved in SULI-mediated degradation*" (Lines 211-214).

7) The new conclusion "Thus, temperature seems to have a significant effect on the performance of LIST in regulating protein stability in different organisms" does not directly follow from the data and should be adjusted.

Response: As suggested by the reviewer, we have adjusted the conclusion in the revised manuscript (Lines 380-382).

8) The new conclusion "It is also possible for LIST to optogenetically control protein stability in mammalian cells after further systematical optimization of the photoinducible characteristics of VVD." also does not directly follow from the data and should be adjusted.

Response: We have removed the description from the manuscript to avoid any confusion, and thank the reviewer for pointing it out.

9) Similar to my previous assessment, reviewer 1 stated that "Moreover, in the western blot shown in Suppl Fig.S9, the levels of mCherry are much higher than those of mCherry-LIST regardless of whether the sample was in the dark or illuminated. This strongly suggests that fusion to LIST dramatically decreases the expression level of the POI". As I had pointed out previously, readers might therefore perceive "light-inducible stabilization tag" as a misnomer since tagging with LIST per se already seems to destabilize fusion partners such as mCherry. However, the authors decided not to use an alternative name.

Response: We have used an alternative name of SULI (stabilization upon light induction) following the reviewer and editor's suggestion.

10) I do not agree with the new phrasing that "It should be aware that the data produced with western blot is typically considered to be semi-quantitative, as there are many factors that can affect the signals of target protein, e.g., antibody affinity, transfer efficiency, protein configuration, linear range and internal loading control 1, 2. Thus, it is not recommended to precisely quantify the levels of target proteins just according to the

signals of the bands, especially when the target proteins have different sequences, molecular weights, and configurations. Instead, determination of the fluorescence intensity is a more straightforward way to precisely quantify the levels of a target protein fused with a fluorescent protein (such as GFP or mCherry), which has been widely used in biological studies 3, 4.". The authors point to the correct guidelines for quantitative western blotting, and relevant experimental data following these guidelines can indeed be found in the literature. On the other hand, it is not correct that "determination of the fluorescence intensity is a more straightforward way to precisely quantify the levels of a target protein fused with a fluorescent protein". In fact, fluorescence intensities can be prone to structural changes; particularly prominent examples are nanobodies that can change fluorescence intensity without affecting protein abundance – here, using fluorescence intensity as a proxy for protein levels would be misleading.

Response: To eliminate the misleading, we have removed the description of using fluorescence intensity as a proxy for protein levels from the legend of Supplementary Fig. 11, and greatly appreciate the reviewer's valuable suggestion.

11) The paper now lists a patent application and deleted "and materials" from the "Data and materials availability" section. The authors should include a statement that the constructs are freely available without any restrictions.

Response: We are pleased to freely share the optogenetic tool developed in this study to all scientists around the world who are interested in applying the tool in their studies for not-for-profit research. As suggested by the reviewer, we have added the description "*The constructs generated in this study are available upon request from the corresponding authors with appropriate Material Transfer Agreement (MTA)*" in Data availability section.